# Process Mining IPTV Customer Eye Gaze Movement Using Discrete-Time Markov Chains †

**Zhi Chen** [1,*], **Shuai Zhang** [1], **Sally McClean** [1], **Fionnuala Hart** [1], **Michael Milliken** [1], **Brahim Allan** [2] and **Ian Kegel** [2]

1 School of Computing, Ulster University, Jordanstown BT37 0QB, UK
2 BT Applied Research, British Telecom, Ipswich IP5 3RE, UK
* Correspondence: chen-z7@ulster.ac.uk
† This paper is an extended version of our paper published in Chen, Z.; Zhang, S.; Mcclean, S.; Lightbody, G.; Milliken, M.; Kegel, I.; Garifullina, A. Using Eye Tracking to Gain Insight into TV Customer Experience by Markov Modelling. In Proceedings of the 2019 IEEE SmartWorld, Ubiquitous Intelligence & Computing, Advanced & Trusted Computing, Scalable Computing & Communications, Cloud & Big Data Computing, Internet of People and Smart City Innovation (SmartWorld/SCALCOM/UIC/ATC/CBDCom/IOP/SCI), Leicester, UK, 19–23 August 2019.

**Abstract:** Human-Computer Interaction (HCI) research has extensively employed eye-tracking technologies in a variety of fields. Meanwhile, the ongoing development of Internet Protocol TV (IPTV) has significantly enriched the TV customer experience, which is of great interest to researchers across academia and industry. A previous study was carried out at the BT Ireland Innovation Centre (BTIIC), where an eye tracker was employed to record user interactions with a Video-on-Demand (VoD) application, the BT Player. This paper is a complementary and subsequent study of the analysis of eye-tracking data in our previously published introductory paper. Here, we propose a method for integrating layout information from the BT Player with mining the process of customer eye movement on the screen, thereby generating HCI and Industry-relevant insights regarding user experience. We incorporate a popular Machine Learning model, a discrete-time Markov Chain (DTMC), into our methodology, as the eye tracker records each gaze movement at a particular frequency, which is a good example of discrete-time sequences. The Markov Model is found suitable for our study, and it helps to reveal characteristics of the gaze movement as well as the user interface (UI) design on the VoD application by interpreting transition matrices, first passage time, proposed 'most likely trajectory' and other Markov properties of the model. Additionally, the study has revealed numerous promising areas for future research. And the code involved in this study is open access on GitHub.

**Keywords:** eye tracking; HCI; discrete-time markov chains; process mining; TV customer experience; IPTV; statistical analysis; machine learning; R; Java

## 1. Introduction

The last 15 years have seen the growth of TV delivered over broadband IP networks (often termed IPTV) from a niche proposition to a product which is commonplace in households worldwide, with on-demand viewing now very much integrated into everyday life [1]. IPTV can be seen as an integration of the Internet, multimedia, communication and other technologies delivered over broadband access networks. It provides households with a variety of interactive services and adapts to the rapid development of networks, making full and effective use of network resources [2]. IPTV typically offers at least the following three features: live broadcast, replay and on-demand viewing. Replay refers to recording programmes that have been broadcast in the past few days so that customers can choose to watch them later with the option to fast-forward or rewind. On-demand viewing is often more interactive and personalised, with customers provided with dedicated applications offering libraries of TV content for viewing at any time.

In the past, the TV experience involved a fixed number of channels showing one programme at a given time, with the customer only having the choice of a limited variety of channels [3]. Today, however, this linear approach has been totally redeveloped with the use of IPTV services. This advancement in technology has not only improved the viewing experience, but it provides more detailed data to analyse, and this, in turn, can ultimately improve the customer's interaction with the service. For instance, we have seen on IPTV platforms how algorithms are used to predict other films or TV shows that the customer may be interested in based on their viewing history. For many, this is a welcome development, as it reduces the need to scroll through significant numbers of pages in an attempt to search for a film, particularly if the user is simply "browsing" and has no specific film in mind to watch. As a consequence, it has been a trending research topic for IPTV service providers or video-sharing platforms to gain insights into users' interaction data regarding their preferences for programme content, habits of using the Electronic Program Guide (EPG), patterns of TV viewing time on a daily/weekly/monthly basis, or quality of the service, etc. In the meantime, analytics technologies have been widely used for data provided by a variety of devices and viewing media for industrial and academic study in order to get a deeper knowledge of TV viewers' behaviour [4]. In the IPTV domain, studies of customer behaviour normally fall into one of the following categories: One is related to how they interact with the TV service plan and Electronic Programme Guide (EPG) provided by the IPTV service provider. Such actions can be briefly described as "subscribe to the service," "turn on the set-top-box," "use the remote control," "watch a programme," "switch channels," "add a programme to favourites," "purchase a programme," "unsubscribe to the package," "leave the service," etc. The other direction of study is more focused on customers' preferences concerning programme content and genre when they are watching IPTV. Respectively, there have been studies related to IPTV customers, including behaviour prediction (e.g., churn prediction [5], channel switch prediction [6,7]) and recommender systems [8].

Since the interactivity of IPTV systems and similar video-content-based platforms has been significantly developed in the past years, it has also brought along challenges for usability studies on IPTV systems to improve the quality of the user interfaces. Besides, the interactions between users and IPTV systems take various shapes depending on the source of data. Ref. [9] has conducted a case study on an IPTV app, testing the usability of the system. The focus of the study is on optimising the UI/UX. They provide a solution to define and evaluate the usability problems of the app by investigating remote control data in terms of the interaction between users and IPTV systems. However, there are fewer studies on customers' engagement with IPTV interfaces based on their eye movements. Given that the data behind those IPTV platforms are already being put to use to improve the user's viewing experience, it follows logically that the companies behind these IPTV services would also try to find other methods to improve and enrich their platform in any way they can. In that regard, eye-tracking technology has been introduced in the study of interactions between TV users and the IPTV system regarding gaze movement. The concept of eye tracking has rapid improvement in recent times with advancements in technology, and therefore so has our analysis of the data. Our eyes analyse the world at an almost constant rate; therefore, it makes sense that this information should be used to improve our daily lives. There are potentially many uses for eye tracking technology in multiple areas, to improve professional performance in the workplace, to aid scientific research, and for the Gaming and Virtual Reality (VR) industry, eye tracking can also vastly improve their research. The study in [10] introduces a list of eye-tracking technology implementation methods, including single and multi-camera eye trackers. Eye tracking can provide insight into behaviours and interactions with on-screen content and the surrounding environment [11]. The process through which humans make observations on multimedia content can be analysed via eye tracking [12], which is relevant for IPTV settings. In addition, ref. [13] presents a method for evaluating interface usability based on eye movement data. Both [14,15] of the more recent studies [14,15] use eye-tracking equipment to analyse viewing behaviours.

Ref. [14] provides a better representation of web pages by combining the user's viewport screenshots with fixed elements on the web page, subsequently assessing the usability of the web page. Ref. [15] utilises eye tracking technology in modelling the eye movement of two different groups of learners reading the same material on the screen to assess their learning experience. In this paper, we focus on the application of eye-tracking technology for the benefit of the Human-Computer Interaction (HCI) study, especially in the IPTV domain.

Previously, at the BT Ireland Innovation Centre (BTIIC) in Belfast [11], an eye tracker (Tobii X60) was used to evaluate the user experience on the BT Player, a video-on-demand application supplied by BT as part of their IPTV service in the United Kingdom. The collaboration between BT and Ulster University through the BT Ireland Innovation Centre (BTIIC) has allowed eye-tracking technology to be applied in a typical user setting of the BT Player application and, subsequently, the data from these experiments can potentially offer additional insights into user participation that have not been accessed before. The contributions of this paper can be generalized as follows. Firstly, the paper presents a follow-up investigation of the experimental results by extending our early-stage work [16]. We here apply a first-order discrete-time Markov Chain (DTMC) [17] to model and describe the data generated by the eye tracker, as well as gain insights about customers' interaction with BT Player in terms of their gaze movement. Our hypothesis is that the gaze movement process of participants watching IPTV satisfies the Markov assumption, which is that what the participant is gazing at now only depends on what s/he last gazed at and does not depend on the rest of the gaze history. We have created a pipeline of data analysis and modelling and come up with interpretations of created DTMC properties for specific scenarios as well as evaluation of user interface when using the BT Player on different pages. Also, we have generated a series of scripts using Java, R Studio, MySQL, etc., and relevant code scripts are now available on GitHub for reference through the link: https://github.com/Benedict1123/Eyetracking, accessed on 18 January 2023. The study in our paper has demonstrated a comprehensive methodology to combine eye-tracking technology with IPTV products, along with machine learning techniques, to generate explainable results for the purposes of evaluating and improving the UI design of the IPTV platform. The main difference between our study and previous research in the eye-tracking and HCI fields is that we focus on the sequence in the gaze movement from the eye-tracker data, and we also introduce time into the methodology when calculating the mean passage time of a DTMC. Previous literature shows that researchers usually carry out quantitative analysis of eye tracker data such as fixation counts, fixation period, etc. The methodology in our study can be applied to other IPTV products or systems with human interactions for further research.

The remainder of the paper is structured as follows. The next section talks about the research background, regarding previous related work on applications of eye tracking technology, human-computer interactions, Markov Modelling, etc. In Section 3, a detailed discussion of our methodology is covered, including the setup of the eye-tracking experiment at BT and the relevant analytical techniques we have applied. Section 4 highlights analytical results, and Section 5 shows our evaluation of the entire study. Finally, we draw some conclusions in Section 6, supported by BT's industrial feedback on the results.

## 2. Related Work

Previous research has documented that eye-tracking technology has been widely used across many areas with regard to customer behaviour studies and production evaluation. Also, the advancement of Human-Computer Interaction (HCI) has developed considerably as we attempt to understand not only how humans interact with computer screens but why we make certain choices or our attention is drawn to specific areas of the screen rather than others. If this can be understood, then it can be manipulated to improve customer engagement with an IPTV service. Moreover, Markov chains have been extensively incorporated

into studies on HCI and other sorts of behavioural data, such as eye gaze movement, which is also known as sequential or time series data.

*2.1. Eye Tracking in Research*

Eye tracking has evolved over many years in terms of both its technology and its applications. In 1980, Just and Carpenter [18] first brought up the concept of the "eye-mind hypothesis," which was related to studies on reading comprehension. Their work was based on the assumption that "the eye remains fixated on a word as long as the word is being processed," which indicates the strong correlation between what a person is looking at and cognitive processes [11]. The first application of eye-tracking technology was to assist severely disabled people with daily activities by designing an eye-wink interface in the 1990s [19]. Since then, eye tracking has been utilised in a variety of fields of research and commercial applications.

For example, one study [12] employs Tobii X60 (a model of eye tracker) to assess how students learn from multimedia sources. In the experiment, there are two groups of students from the second and sixth grades of Primary Education, and they are presented with four different sets of multimedia presentations that include verbal content. Their eye movements are recorded by the eye tracker and measured by the pre-defined metric named "learning efficiency." As a result, they validate the hypothesis that different formats of materials (e.g., text only, image with text, image with audio) work differently for the two age groups. Similarly, the research [15] on the difference between English as a foreign language (EFL) beginners and intermediate-level readers when they read narratives comprised of text and images also applies eye-tracking technology to the analysis of eye movement on the target areas of interest (AOIs). In the same area, to help assess school-aged children's virtual attention and identify potential risks of learning or attention problems, the authors [20] propose an integrated platform consisting of a GP3 eye tracker and the OGAMA application, etc. The platform shows some great potential for future applications in education, VR, or AR development. The same eye tracker and software set were employed in another study in the education area [21]. The authors set up an experiment by recording volunteers' eye gaze moments while they are doing code debugging in front of the computer screen. The study comes up with some algorithms to calculate the efficiency of debugging between two groups of participants. The characteristic parameters are taken into account when dividing participants, and the results show a significant difference between the two groups in terms of efficiency and the number of fixations. It reveals a more efficient way of doing software development. Also, the study has set a successful example for measuring people's perception abilities in the field of education.

Eye-tracking technology has also played an important role in the research area of e-commerce. For example, ref. [22] has proposed a framework for evaluating the performance of an e-commerce website in terms of item recommendation. They introduce a neural network at the heart of the framework that is trained with synthetic data regarding the user's demographics and other characteristics. And a GP3 eye tracker is the main piece of equipment in the study to provide gaze movement data on the website with both a vertical and horizontal layout. The results of the study have demonstrated the high prediction accuracy of items being added to baskets and also provided a solid solution to automating an optimal recommendation interface adjustment for an e-commerce website. Additionally, ref. [13] has proposed a new method of evaluating webpage usability by the analysis of eye movement. Eye movements are recorded using a head-mounted eye tracker when participants are instructed to look at four webpages twice per page and locate relevant information. By comparing all participants' records of performance measurements (i.e., response score, task completion time) with eye movement measures, the study indicates that eye movement data may be used in conjunction with performance measures to acquire a deeper knowledge of usability issues. Similarly, ref. [14] employs eye tracking to estimate user attentiveness and evaluate the usability of website pages.

Eye tracking has also had success in lots of different areas of consumer research for a variety of reasons. In traditional research, a task may be carried out by the participant and then afterwards, they would be asked questions based on that task. For example, in a supermarket, a customer could be asked to carry out their daily shopping and then afterwards asked if they noticed certain labels, offers etc. and why those choices may have seemed appealing [23]. One of the problems within this area of research is that human memory is not faultless, and sometimes the customer will either forget, not notice or not think about why such a decision was made, which can make giving an answer to some of these types of questions difficult, or can provide false information at times [24–26]. The benefit of eye tracking software is that it has the ability to capture the gaze position of the participants' eyes, and this position will be recorded at different time intervals depending on the device. This removes the potential for any memory issues and therefore removes human error in the process. If the trajectories of the participants' eyes can be monitored at all times, then the reasons behind their choices become more apparent, whether because of colour, a bolder font, or a special offer price [27–29]. The purpose of the study at BTIIC was to evaluate TV customer experience by measuring participants' eye movements while using BT Player on a television [11].

The BT Player screen is very visual, showcasing available material via a combination of menu items, controls, and images (See Figure 1 below). The eye tracker was utilised to document the user's engagement with the interface. During the step of data extraction following the session with participants, the recorded screens for each task were separated into AOIs. Fixation occurs when an AOI is selected, but saccades occur between each AOI. A trajectory is the path of gaze travel between numerous AOIs. Understanding and constructing such a model of user behaviour helps facilitate prediction.

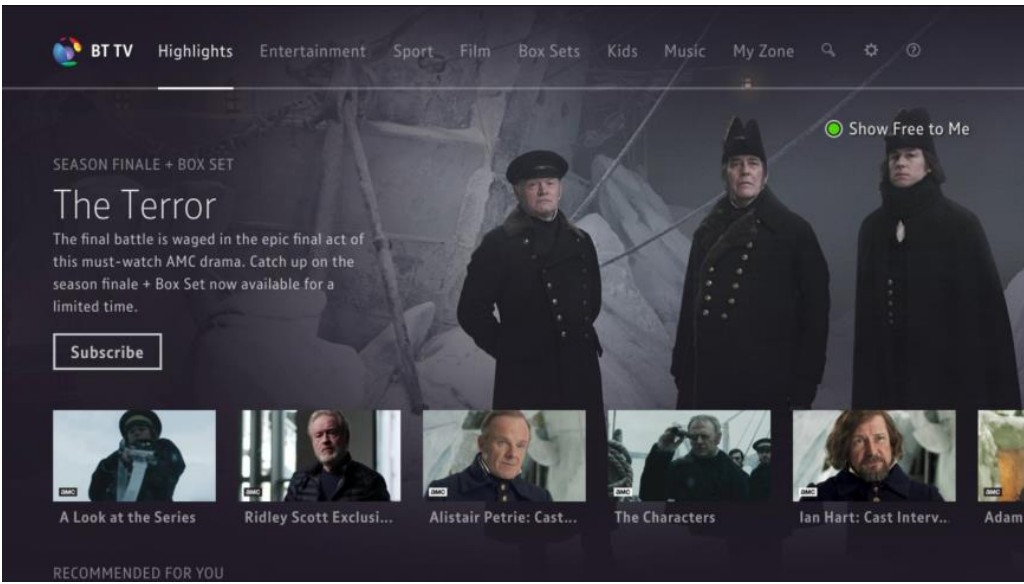

**Figure 1.** BT Player Screenshot [16].

*2.2. Human-Computer Interaction*

Advancement in technology and the use of computer screens in everyday life has rapidly expanded the research into Human-Computer Interaction (HCI) and how users explore the screen, both with their eyes and where they point or click on the screen. This research has provided useful information for graphic designers who want to make websites as efficient and user-friendly as possible while also conveying the most crucial information clearly. It is necessary, therefore, to understand why the human eye is attracted to certain regions or objects on the screen to be able to adapt these websites accordingly.

2.2.1. Fitts' Law

Fitts' Law can be used to understand why human behaviour demonstrates certain distinct patterns, both on computer screens and otherwise. This law states that the time required to point at an object using some form of a pointer (either on a computer screen or otherwise) is a function of the distance from the desired object to be pointed at and the size of that object. Leading on from that, it suggests that if the object is both small and far away, then it will require a longer time for someone to locate and point to the object than if it is closer and larger, as can be seen in Figure 2 [30], where the graph shows how the object size interacts with the Usability Index of that object. The equation to define Fitts' Law is:

$$T = A + Blog_2\left(2\frac{D}{W}\right) \tag{1}$$

where:

- $T$ is the Time required to point to the object;
- $A$ and $B$ are empirically determined regression coefficients;
- $D$ is the distance from the pointer to the object;
- $W$ is the width of the object text following an equation, which need not be a new paragraph.

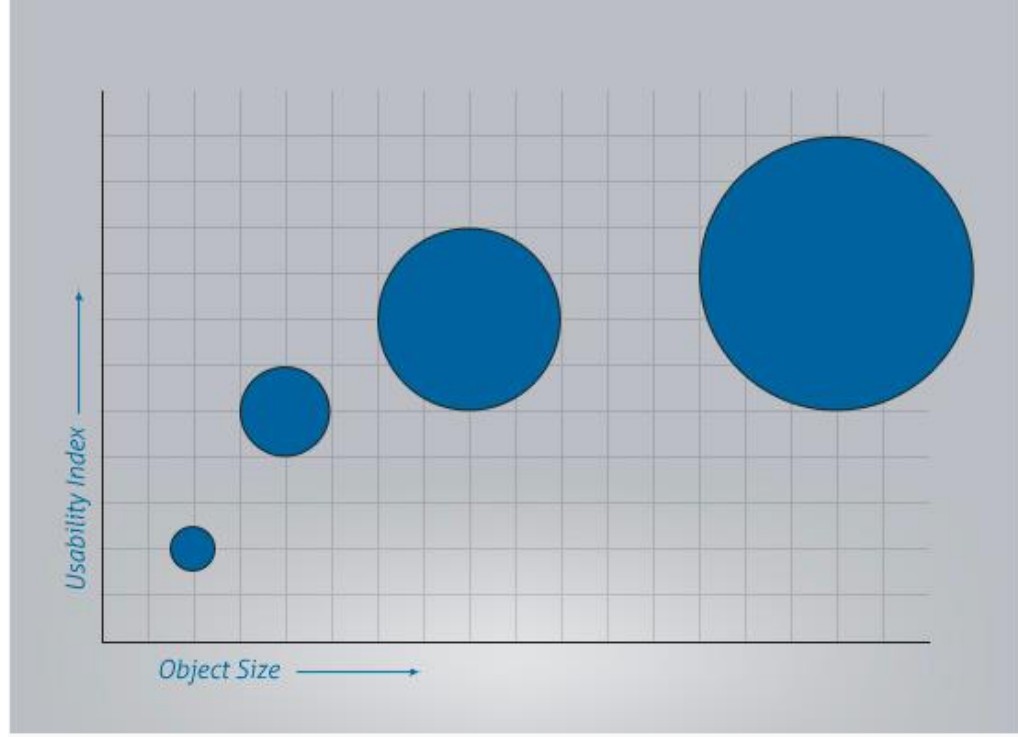

**Figure 2.** Fitts' Law Graph–Object Size vs Usability Index [31].

Although Fitts' Law was initially established in 1954, it has developed great significance in recent times due to the desire to improve human-computer interaction and the ergonomics of the computer screen. Businesses want to promote certain aspects more than others, such as more expensive items, or make certain buttons clearer and more defined, e.g., the 'Continue' button rather than the 'Back' one. Using Fitts' principles, it has become clearer to graphical designers that command buttons should be larger and distinguishable from other features on the page. Additionally, when smaller objects are far apart, it takes longer for users to select than larger ones that are closer together, so a pie chart menu is preferable to a drop-down menu or a horizontal/vertical task bar etc. All of these features have been taken into account in recent graphical design, with the basis of Fitts' Law [31,32].

### 2.2.2. Gestalt Principles

Taking Fitts' Law a step further is the Gestalt Principles. Gestalt is German for "shape," and the principles look at proximity, similarity, continuity and closure between objects and analyse how the eyes may deal with different examples of these. These principles originated in the 1920s from Gestalt Psychologists, who proposed that humans naturally observe objects in certain patterns and in an organised manner, as the brain naturally perceives objects in this way [33–35].

- Proximity: This principle states that if objects are within close proximity to each other, the brain naturally groups them compared to those that are further apart;
- Similarity: This principle suggests that the brain groups objects based on their similarity, in relation to colour and shape etc., and distinguishes those that are different as a separate group;
- Continuity: The brain naturally follows and continues lines, even those that intersect with each other, and forms groups based on this continuation;
- Closure: In relation to shapes, if the brain observes lines which form incomplete outlines of certain shapes, we naturally close the gaps to form that shape as the brain prefers completeness and, therefore, initially views the shape as a whole.

### 2.2.3. F-Shape and Horizontal Left Patterns

A study in human-computer interaction using eye-tracking technology has also been used to better understand human on-screen behaviour, out of which two clear patterns have emerged-the "F-shape" [36] and "horizontal-left" [37]. The terms come from gaze pattern analysis which has shown that the human eye naturally reads material in an F-shape, reading horizontally from left to right along the upper content of the page, then travelling vertically down the left-hand side and traversing horizontally across the next section of information again from left to right, but not continuing along as far to the right-hand side as the first section, before finally continuing vertically along the left-hand side before finishing. This pattern forms an "F" shape on a heatmap produced by the gaze pattern [36].

In terms of the "horizontal left" pattern, this refers to the user's eye-tracking patterns appearing significantly more on the left-hand side of the page than on the right. A study performed in 2010 found that 80% of fixations fell on the left-hand side of the page, with only 20% of fixations on the right-hand side [37]. Taking the previous information regarding the typical "F" shape reading pattern, it is understandable that these figures are so disproportionately high as the scrolling vertically on the left-hand size would lean to a higher amount of left-side fixations. With this information, graphic designers are able to adjust the layout and content on websites accordingly, placing the most important information in the top few lines and along the left-hand side of the page where viewers are most likely to fixate on it [36].

### 2.2.4. HCI Evaluation Techniques

There already exist multiple methods to evaluate HCI systems, which differ across computer and television platforms. However, IPTV may require a combination of these techniques as it involves interactive television screens in different environments to traditional computer use [38]. Methods such as Cognitive Walkthrough, which involves an expert evaluating the HCI design to identify any potential problems [39], or Think Aloud/Cooperative Evaluation, where the user is observed performing a task and is asked to comment or collaborate on the evaluation, have been used to evaluate and improve HCI designs [40]. These traditional methods have previously been useful for evaluating existing screens; however, with the emergence of IPTV services, a new approach could be taken to evaluate the interactive element of these designs.

In the field of e-commerce, which was also introduced in the literature review in the previous part, research on human-computer interaction is also very popular. Many of them study the behaviour of users during the shopping process on the website, analysing how they browse products and finally add them to the shopping cart. This type of research

can help websites better understand user preferences and the impact of website design on user transactions. In one of the studies [41], the authors developed a tool to analyse users' shopping behaviour on e-commerce websites from the perspective of human-computer interaction. This tool can combine a series of user-triggered front-end DOM (document object model) events, as well as mouse and keyboard input, and conduct experiments on product recommendation systems with three different page structures, comparing the impact of web page structure on user browsing and impact on product purchases. They concluded that vertically structured product areas are more attractive to users than horizontally structured product areas. Their future research plan is to introduce eye-tracking technology to add the user's eye movement trajectory and combine the current DOM information to generate a more accurate user interaction behaviour model.

There are other types of human-computer interaction, generally depending on which device a person uses to operate and control a computer system. Among the game consoles we are familiar with, the Xbox 360 has launched a Kinect camera to capture the user's body movements to interact with the game. The Wii game console provides a somatosensory handle to capture the waving movements of the user's hands and arms and then operate the game character. In another study [42], the authors used an IoT device (a Leap Motion controller), which can capture the movement of the user's palm and fingers, allowing the user to use the hand instead of the traditional mouse to remotely operate the computer. The study invited 32 volunteers to participate in the experiment. The content of the experiment was to use the Leap Motion controller to complete an IT task on the computer. At the same time, they also asked the volunteers to use a traditional mouse for the experiment as a control. The researchers analysed the movement of the cursor on the screen of the traditional mouse and the Leap Motion controller and carried out qualitative and quantitative analyses and comparisons. Similar to the previous research methods using eye-tracking technology, they also use indicators such as "fixation number" and "fixation period." Additionally, the field of human-computer interaction also includes research on the interaction between the human brain and machines because, fundamentally speaking, human interaction behaviours are actions taken after receiving brain signals. For example, in an analysis based on Electroencephalograph(EEG) and user behaviour [43], the author used a brain wave acquisition device (the MindFlex EGG headset) to collect brain activity information from volunteers and developed a system to process and quantify brain waves, including graphical visualisation of brain signals over time. Their research results can be redeveloped and applied to other human-machine control research.

Based on the literature above, eye-tracking technology has been chosen for this study as a method to evaluate these designs; however, to date, it has largely been used to monitor television or video/computer screens, where there is fixed content that all users watch and then the Areas of Interest (AOI's) can be analysed. For IPTV, the challenge with eye tracking is that users have much more choice in the content that they choose, meaning users can dynamically change the screen. This can potentially give significant insight into the use of this service, while the analysis can be more complex as a 'one-size-fits-all' approach will not work in terms of annotating AOIs, which may be different for each user and at different points in time.

### 2.3. Markov Chain Application in Process Mining

A Markov chain or Markov process is a stochastic model describing a series of events where the probability of each event is dependent on the state attained in the previous event [44]. And there are overall two types of Markov chains according to the interval type of the change between states in the chain, discrete-time Markov Chain (DTMC) and continuous-time Markov Chain (CTMC). They are appropriate for modelling data which contain a random process and the state of the system changing over time. Therefore, Markov chains have been widely applied to the exploratory analysis of time-series (i.e., sequential) data over a large range of domains.

For instance, a kernel variable length Markov Model (KVLMM) is proposed in a study about GPS trajectory prediction [45]. Similarly, in the context of social robotics, there has been a study [46] using machine learning techniques as well as eye trackers, specifically incorporating Markov Models for the stochastic process of gaze movement. The authors have developed a probabilistic algorithm for gaze trajectory prediction. In other industries, such as communication, a study in Nigeria [5] applies Markov models to the telecommunication industry use case, helping to predict customers' churn and retention rates. For bank customer behaviour modelling, ref. [47] proposes Markov chains to describe customers' dynamically changing delinquent status and behavioural scores. In [48,49], clickstream data are explored for studies on online purchasing processes, and a Markov Model is used to characterise page changes. Similarly, refs. [50,51] utilise Markov Models to visualise website clickstream data. Another study [52] uses a Markov Model to construct an intelligent pre-loading system for online resources, wherein the navigation graph is considered a Markov model. In addition, studies on healthcare data [53] emphasise the use of non-homogeneous Markov Models for sequential pattern mining of records pertaining to the behaviours of elderly patients.

Meanwhile, the trending application of Markov chains has also facilitated the development of handy packages for modelling. There are R packages (such as 'clickstream' [48] and 'markovchain' [17]) based on Markov characteristics that facilitate study on applications of the Markov Model. The 'mar-kovchain' package closes a gap in the R framework for discrete and continuous time Markov chain analysis as well as probabilistic statistical analysis [17].

Above all, in this paper for our study, compared to previous studies in terms of the eye tracking application and HCI analysis techniques, here we focus more on the trajectory of gaze movement, which is about the gaze transition information. Also, to distinguish our method from traditional quantitative analysis of eye tracker data, we chose Markov chains to model the viewer's gaze movement when they're watching the BT Player.

## 3. Methodology

As previously mentioned, this paper is subsequent work to the previous eye-tracking experiment at BT and Ulster University [11,16], in which a group of participants used BT Player while their eye movements were monitored by an eye tracker. The dataflow of the eye gaze movement is integrated with other information, such as the BT Player screens, by the Tobii Studio (version 1803, Windows) software package. For the duration of the experiment, participants were instructed to complete several activities connected to TV viewing. This study focuses largely on 2 activities, namely, how TV customers acquire content and how their eyes stay on the Content Discovery page for the first few minutes throughout this activity. Besides, since Markov models have been chosen to model the eye-tracking data, this section discusses how the calculations and techniques can be implemented. As with most forms of data analysis, certain date manipulation needs to take place to achieve the required format before the Markov analysis can begin.

### 3.1. Experiment Design

As part of a collaboration between BT and Ulster University, an eye-tracking experiment was undertaken, with participants using the BT Player app while a Tobii eye-tracking device monitors the gaze positions of each participant's eyes. This experiment was set up to develop a richer understanding of how customers use BT Player, which could potentially be used to improve the design and usability of future versions of BT's TV service [11]. An overview of the experiment setup can be seen in Figure 3 below, which shows how the TV screen and eye-tracking device were set up for each participant [11]. Each participant must be sitting the same distance from the screen, with the eye-tracking device positioned near them on an adjustable stand to account for different participant heights, so the eye-tracker can be a fixed distance from their eyes, as the maximum gaze angle must be 35 degrees. All of these measurements must be fixed across the experiment to ensure consistent results.

Data from the eye-tracking software has been collected by Ulster University during an experiment where 14 individuals each spent a period using BT Player while being monitored by eye-tracking software. Each individual completed a survey asking for demographic information such as gender, age, and familiarity with BT Player and other popular TV services. The individuals were required to complete a series of tasks throughout this period, which were based on the requirements provided by BT.

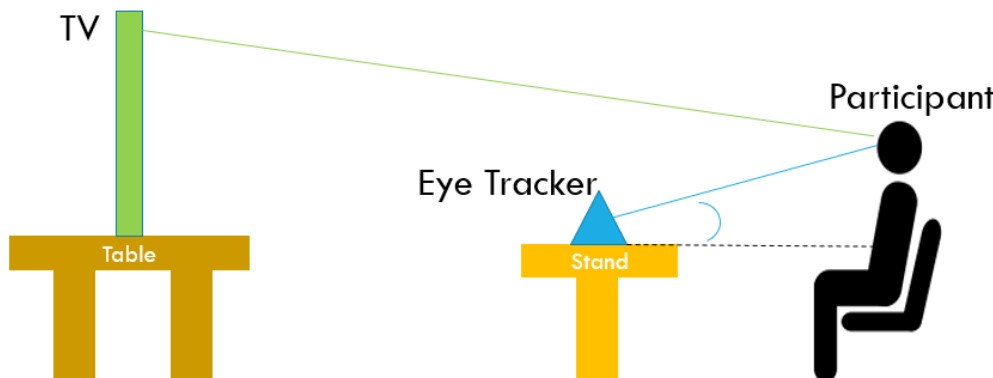

**Figure 3.** A complete view of the test environment [11].

### 3.2. Aims and Objectives

For the whole eye tracking study at BT and Ulster University, BT initially provided a list of requirements of interest, intending to improve BT Player from the perspective of UI/UX evaluation. In this paper, we conduct our research by investigating 2 of them. These requirements targeted on specific pages on BT Player are listed below, with a brief explanation of each:

1.  Purchase Flow Pages: BT is interested in how the user interacts with the TV on Demand service to improve the ease of use of the purchase flow (from initially choosing a TV show/film to going through with payment) to ultimately increase sales;

2.  Content Discovery Pages: BT is interested in how the user interacts with the main pages of the BT Player, regarding searching for items, looking at menus and carousels (large images and descriptions on the screen to draw attention), to improve the user interface of these pages to increase sales.

More specifically, there are comparatively representative sub-requirements under each requirement. Thus, we focus on one specific sub-requirement for each high-level one shown above as our research objectives defined by BT:

3.  Content Purchase: "When purchasing content (TV on Demand), what draws the eye? Is it the price, is it the quality, or is it something else?"

4.  Content Viewing: "When a Content Discovery page first loads, what are customers viewing? Are they drawn to the hero carousel, the navigation or something else?"

In the study of a "content purchase" scenario using the BT Player, the eye-tracking data taken from the experiment will be analysed. With the use of the eye tracker software (Tobii Studio), various Areas of Interest (AOIs) can be highlighted on each screen, so the Purchase Flow screens can be analysed, and the AOIs chosen as certain sections or buttons on that screen that are more of interest than others. For example, the buttons to select a specific payment option, as well as any images or information relating to the content that is available on the screen. In general, the eye tracker records data about the user's gaze point on the screen, in addition to other relevant variables. This data will be analysed using mathematical modelling techniques to determine if certain features on the Purchase Flow pages of BT Player are of significant interest or if the user does not appear to notice other features. Likewise, the same methodology is also applied to the "content viewing" scenario, which is aimed at providing some insights into the gaze movement on the Content

Discovery page in terms of customers' navigation paths, thereby generating feedback on how the user interface could be configured to be more accessible for users. Previously, there was some work related to this scenario published in October 2019 [16].

Ultimately, our study aims to use eye-tracking technology to gain a better understanding of how users interact with BT Player in relation to the purchase of films and TV shows, as well as the discovery of content available throughout BT Player in terms of carousels, menus etc. The trajectory of gaze movement is analysed while participants are fixating on the purchase screens to highlight the areas on the screen that appear to be more of interest to the participants and attempt to gain some insights into why this is the case. Moreover, the analysis of gaze movement on the content discovery is to demonstrate the top eye-catching parts among participants, along with some preferences on the order of viewings, so that designated parts on the screen can be highlighted accordingly to draw more attention for various purposes. Consequently, the underlying process of the customer's eyes traversing over the screen is revealed.

### 3.3. Data Manipulation

#### 3.3.1. Data Collection

Tobii Studio is utilised to capture gaze data from an eye tracker. As the entire experiment comprised numerous tasks, it was required to partition the entire data set and obtain the relevant information for the evaluated task. The target dataset was afterwards exported as '.csv' files. MongoDB was used to store the dataset in order to gain an overview of the data structure prior to beginning the study due to the dataset's raw data format and massive information on its numerous features. Also, the Tobii software (version 1803, Windows) allows the user to define certain AOIs on the screen concerning sections of the screen which may be of more interest than others. Thus, we had a list of pre-defined AOIs on BT Player during the eye-tracking experiment.

#### 3.3.2. Data Pre-Processing

The raw data from Tobii software provides a significant amount of information by default, while only some of which are necessary for creating a Markov Model. It originally came in JSON format, and Tobii Studio converted it into CSV files. We saved the raw data into a MongoDB table to get a quick overview of it. There were only six attributes in the MongoDB database that were relevant to our research: "id eyetracker," "participant name," "local timestamp," "GazePointX (ADCSpx)," "GazePointY (ADCSpx)," and "gaze event kind." Initially, there were 59 columns in the table. The six variables extracted are described below:

- id eye-tracker-time sequential ordered list based on the timestamp where each recording was taken (i.e., the first recording is 1, the second is 2 etc.);
- participant name-14 participants (P001–P014);
- local timestamp-timestamp taken every 00:00:00.165 s (i.e., 10:07:46.441);
- GazePointX (ADCSpx)-the co-ordinates of the gaze-point in the X-direction;
- GazePointY (ADCSpx)-the co-ordinates of the gaze-point in the Y-direction;
- gaze event type-can be "Fixation," "Saccade," or "Unclassified."

Those variables are extracted for each participant, and all of the data is combined in one database using a JavaScript program before the coordinates can be defined as AOIs. As the participants were looking at different parts of the BT Player app during the experiment, it is only the times during which they were on the Purchase Flow and Content Discovery pages that are of interest; therefore, the recordings of each participant in the study were reviewed, and the timestamps based on when the participant was on relevant pages were recorded.

However, some of the columns pertaining to the aforementioned key properties originally contained blank values. This is because the eye tracker continues to record throughout the whole trial, including the participants' eye movements when they glance away from the TV screen. The only valid region for the eye tracker in this experimental

configuration is the TV screen. Once the gaze point falls outside of the barrier, the columns "GazePointX (ADCSpx)" and "GazePointY (ADCSpx)L" will have null values. A Java application is used to assign zero values (i.e., (0,0)) to certain types of records in order to maintain their status as objective data in this study. The application connects to the MongoDB database and replaces blank values in target columns with zeros in a recursive manner. By extracting information from these six important columns, irrelevant attributes from the original dataset are deleted. A MySQL database is created to store the data in order to maintain a well-structured data schema and conduct queries using rows and columns as opposed to JSON-formatted data (e.g., csv files). To do this, a Java program is written that can connect to both the MongoDB and MySQL databases.

### *3.4. Fitting into a DTMC Model*

3.4.1. Markov Definitions

Considering that the eye tracker records gaze movement at a particular frequency, which makes it a discrete-time process, discrete-time Markov chains (DTMC) are chosen to model the trajectory of participants' gaze movement on the screen in our study. Before going deep into the modelling approach, we would like to provide a brief description of several commonly discussed terms in a DTMC and relevant properties which are primarily used in our study.

- DTMC

The concept of a Markov Model uses a system with a set of 'states' $S = \{S_1, S_2, \ldots, S_n\}$ where n is the number of states in the system, so for example, if $S = \{A, B, C, D, E, F \ldots \}$ $S_1 = A$, $S_2 = B$, etc. [54]. In a first-order DTMC, the probability of the current state is based solely on the previous state of the Markov chain. That is, where the time instants associated with state changes are $t = 0, 1, 2, 3 \ldots$ and the actual state at a given time $t$ ($t \geq 1$) is denoted as $q_t$, the probability of arriving at a given state, given its previous state, can be calculated as follows:

$$P(q_t = S_j \,|\, q_{t-1} = S_i) \tag{2}$$

- Dependency Test

For successive events to form a Markov chain, these events must be dependent on each other, which can be tested statistically [55]. Let $p_{ij}$ be the probability that the system moves from state $S_i$ to state $S_j$ in one step. For successive events to be independent in a first-order Markov chain, the statistic $\alpha$, is defined as:

$$\alpha = 2 \sum_{i,j}^{k} n_{ij} ln \frac{p_{ij}}{p_j} \tag{3}$$

is distributed asymptotically as $\chi^2$, with $(k-1)^2$ degrees of freedom (DF), where $k$ is the total number of states. The marginal probabilities, $p_j$, can be calculated as:

$$p_j = \frac{\sum_i^m n_{ij}}{\sum_{i,j}^m n_{ij}} \tag{4}$$

where $n_{ij}$ is the frequency of transitions from state $i$ to state $j$ [56].

- Transition Matrix

Given that these processes can be considered independent of time (as they are discrete with equal time intervals, a set of state transition probabilities $a_{ij}$ can be calculated by:

$$a_{ij} = P(q_t = S_j | q_{t-1} = S_i), \quad 1 \leq i, j \leq N \tag{5}$$

with the state transition coefficients having the following properties: $a_{ij} \geq 0$, $\sum_{j=1}^{N} a_{ij} = 1$.

This means a transition matrix containing the sets of $a_{ij}$ can be formed with the probabilities where each element of position $(i, j)$ in the matrix stands for the transition probability $a_{ij}$, meaning in each row that a given state $q_{t-1}$ will go to the next state $q_t$. And each row will sum to 1 [54].

- Classification of States

In a Markov Chain, two states, $i$ and $j$, are said to be communicating if they are accessible from each other. In other words, if the probabilities in the transition matrix $a_{ij} > 0$ and $a_{ji} > 0$, then these states are said to be communicating. If all states communicate with each other in the Markov chain, then it is said to be irreducible, and it has only one communicating class, while Markov chains with multiple communicating classes are said to be reducible.

Moreover, a state is said to be recurrent if, upon leaving the state, the chain is certain to return to that state (with a probability of one); however, if the probability of returning to that state is less than one, the state is said to be transient [57]. This can be defined as:

$$f_{ii} = P(X_n = i, \; for \; some \; n \; \geq 1 | X_0 = i) \tag{6}$$

and state $i$ will be recurrent if $f_{ii} = 1$. However, it will be transient if $f_{ii} < 1$.

If a state exists such that when the state is entered, it is then impossible to leave, and the probability of remaining within that state is equal to one. This is known as an absorbing state, denoted in the transition matrix as $a_{ii} = 1$.

- Distribution of States

In each step, $t = 0, 1, 2, 3 \ldots$ , the distribution over all states in the whole system can be represented by a row vector $x$ ($\sum_i x_i = 1, x_i \geq 0$), and the state is denoted by $X_t$. When $t = 0$, we call $x$ an initial state vector, denoted by $\pi_{init}$:

$$\pi_{init} = [P(X_0 = S_1), P(X_0 = S_2), \ldots, P(X_0 = S_n)] \tag{7}$$

In a DTMC, let $A$ be the transition matrix, and $\pi_{init}$ be the initial state vector, then the probability distribution of $X_n$ after $n$ steps is $\pi_n$:

$$\pi_n = \pi_{init} A^n \tag{8}$$

In a Markov Model that is finite, irreducible and aperiodic, an equilibrium can be reached after a long-run period known as the 'Steady State'. This is a vector $\pi$ for a transition matrix $A$ such that:

$$\pi = \pi A \tag{9}$$

which can be solved analytically or through computation. Alternatively, to calculate the stationary distribution over time, $A^n$ can be calculated as $n \to \infty$.

- Trajectory

A trajectory of a Markov Chain is a particular set of values for $X_0, X_1, X_2, \ldots$ . Generally, if we refer to the trajectory $\{S_1, S_2, S_3, \ldots \}$, we mean that $X_0 = S_1, X_1 = S_2, X_2 = S_3, \ldots$ In this study, the trajectory refers to the path of people's gaze movements on the TV screen. Based on the Markov Property, if the transition matrix $A = (a_{ij})$ is known, we can find the probability of any trajectory $\{s_1, s_2, s_3, \ldots, s_{n-1}, s_n\}$ by multiplying together the starting distribution and all subsequent single-step probabilities. The calculation is shown:

$$\begin{aligned} P(s_0, s_1, s_2, \ldots, s_n) \;\; &= P(X_0 = s_0, \; X_1 = s_1, \; X_2 = s_2, \ldots, X_n = s_n) \\ &= P(X_0 = s_0) \times a_{01} \times a_{12} \times a_{n-2,n-1} \times a_{n-1,n} \end{aligned} \tag{10}$$

Therefore, in this case study, we provide a method for determining the probability of the "most likely trajectory" (the trajectory is formed of the most probable condition at each

stage). According to Equation (8), in the $n^{\text{th}}$ step, let $\pi_n$ denote the probability distribution over the states, $s_n{}^{max}$ denotes the state of the highest probability, then we have:

$$P(X_n = s_n{}^{max}) = \operatorname{argmax}\{\pi_n\} \tag{11}$$

Let $p^{max}$ denote the probability of the 'most likely trajectory' in $n$ steps, then can calculate it as follows:

$$p^{max} = \prod_{i=0}^{n} \operatorname{argmax}\{\pi_i\} \tag{12}$$

Meanwhile, the state $s_n{}^{max}$ in each step $n$ comprises a set of states $S^{max}$, which represents the sequence of states from $X_0$. Thus, $S^{max}$ is the most likely trajectory shown below:

$$S^{max} = \{s_0{}^{max}, s_1{}^{max}, s_2{}^{max}, \dots, s_n{}^{max}\} \tag{13}$$

Similarly, ref. [29] introduces the same concept about maximum likelihood trajectories but calculates them in a different way.

- First Passage Time

First passage time [30] is also named hitting time, referring to the time to reach a certain state (if the states communicate). Given the initial state $X_0 = S_i$, the total number $T_{ij}$ of steps taken by the Markov Chain from state $S_i$ to reach state $S_j$ for the first time is the first passage time from $S_i$ to $S_j$. The commonly used quantity related to the first passage time is the mean first passage time. Let $T_{ij}$ be the first passage time from state $S_i$ to $S_j$, so we define $m_{ij}$ as the mean first passage time correspondingly, which is represented as follows [31]:

$$m_{ij} = \mathbb{E}\left[T_{ij} \middle| X_0 = S_i\right] \tag{14}$$

- Transition Matrix Segmentation

If state $j$ is an absorbing state such that it has $m$ transient states and $n$ absorbing states, then the transition matrix $A$ can be separated into sections $Q$, $R$, 0 and $I_n$ written as [58]:

$$A = \begin{bmatrix} Q & R \\ 0 & I_n \end{bmatrix} \tag{15}$$

where:

- $Q$ is an $m \times m$ matrix;
- $R$ is an $m \times n$ matrix;
- 0 is an $n \times m$ matrix of zeros;
- $I_n$ is an $n \times n$ identity matrix [59–62].

This segmentation of the transition matrix of an absorbing chain can allow for further calculations, including the Fundamental Matrix $N = \{n_{ij}\}$, such that the element $n_{ij}$ for N provides the expected number of times the process is in transient state $S_j$, given that the chain began in transient state $S_i$. The Fundamental Matrix $N$ is the inverse of $(I - Q)$ and can therefore be calculated by the equation as follows [59–62]:

$$N = (I - Q)^{-1} \tag{16}$$

- Expected Time to Absorption

Furthermore, if we let $t_i$ represent the expected number of steps before the Markov chain is absorbed, given that the chain began in state $i$, then the column vector $t$ (with an $i$th entry of $t_i$) can be calculated as:

$$t = Nc \tag{17}$$

where $c$ is a column vector of 1 s, e.g., if $N$ is a $3 \times 3$ matrix, then $c$ will be a $3 \times 1$ matrix:

$$c = \begin{bmatrix} 1 \\ 1 \\ 1 \end{bmatrix}$$

Given that this is representative of a discrete-time Markov chain with equal time steps between state changes, if $t$ is a column vector representing the expected number of steps before the state is absorbed from any state $i$, then the expected time to absorption can be calculated by multiplying $t$ by the value of the interval between each state change.

### 3.4.2. Markov Packages–R and MATLAB

The R programming language contributes significantly to scientific research and analysis. This study handles Markov Chain Modelling using the R package "markovchain" [17]. The package contains versatile methods for creating a DTMC object based on the transition matrix, as well as robust algorithms for generating exhaustive results. Assuming that we already have the transition matrix $A$ and a list of states $L$ (state space in the Markov Chain) in R studio, Table 1 below lists the R package functions that we can utilize for our study. In addition, the package can aid in the visualisation of the DTMC model using the R *plot()* function.

**Table 1.** Main functions available in "markovchain" R package.

| R Statement | Function Description |
| --- | --- |
| R > dtmc <- new("markovchain," transitionMatrix = $A$, states = $L$) | Create an object of the "markovchain" class, and, e.g., name it "dtmc" as an R variable |
| R > summary(dtmc) | Display properties and classification of states |
| R > communicatingClasses(dtmc) | Display communicating states |
| R > absorbingStates(dtmc) | Display absorbing states |
| R > steadyStates(dtmc) | Generate the steady-state vector (see Equation (9)) |
| R > meanFirstPassageTime(dtmc) | Create a matrix for the mean first passage times |

In addition, to create discrete-time Markov Models, MATLAB also has an inbuilt function 'dtmc' (discrete-time Markov Chain), where, given a transition matrix $A$ and we let the state names be {1,2,3,4}, the Markov Chain can be constructed using the command: [63] "*mc = dtmc(A, 'StateNames', ["1," "2," "3," "4"])*".

This chain can then be observed as a directed graph to clearly show which states are communicating, if any states are absorbing or if there exist multiple state systems in the chain by executing the "graphplot" function. The MATLAB command is as follows: "*graphplot(mc, 'ColorEdges', true)*". The 'ColorEdges' parameter allows the transition probabilities to be viewed as different colours on a scale of 0–1.

Furthermore, MATLAB can calculate the steady-state vector (xFix) of the chain and the mixing time (tMix), which is defined as the time until the Markov chain is close to reaching its steady-state distribution by the following command [64]: "*[xFix, tMix] = asymptotics(mc)*". Besides, MATLAB also provides solutions to calculate the expected time to reach every state (mean first passage time).

### 3.4.3. DTMC Modelling Steps

In our study, to create a discrete-time Markov Chain (DTMC), there are three primary elements to define and calculate: (1) state space, (2) initial state probability distribution, and (3) transition matrix.

1.　State space–AOI categories

As mentioned before, Tobii Studio allows the user to define certain AOIs on the screen which are of more interest than others. In this case, for both "content purchase" and "content viewing" scenarios, we have divided the screen into several AOIs, represented

by rectangular blocks in Tobii Studio. For the Markov model about gaze movement, those AOIs are the states indicating where the gaze point falls during the Markov process. Similarly, in both scenarios, the onscreen area, apart from defined AOIs, is counted as a whole non-AOI part which is also considered a separate state in the Markov model.

For the "content purchase" scenario, there are two possible screens: A and B. The AOIs for the two different types of screens available in the BT TV Player can be seen in Figure 4 below, with their AOIs defined as either A-F for screen A or A-D for screen B.

Depending on the options available for the video content, screen A has 'Rent HD/SD' options in addition to 'Buy HD/SD', which is why there are more defined AOI E and F than screen B. For both screens, the letter Z has been assigned to the non-AOI sections of the screen. It refers to any area being fixated on the screen which does not include the AOIs A-F or A-D listed above. For example, if the TV viewer is reading the "Buy" or "Buy or Rent?" text information or fixating on the "Cancel" button, this is categorised as AOI 'Z'. For other common AOIs on both screens, A refers to the packshot image of the video, B shows a brief description, and C and D are the buying options for different definitions of quality.

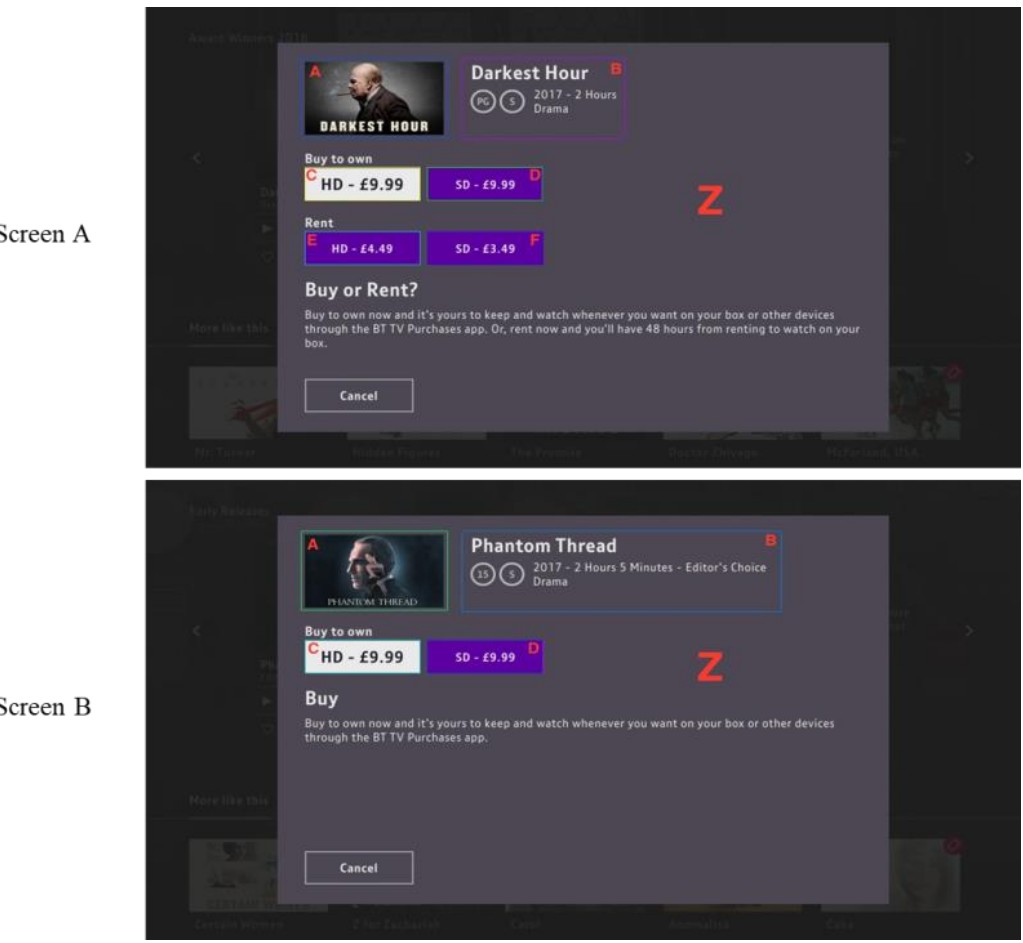

**Figure 4.** "Content purchase"–Screen **A** & **B** (**A**: with rent option; **B**: without rent option).

For the "content viewing" scenario, there is one screen in our study. Similarly, we have divided the screen into AOIs by content. The screenshot of all AOIs on the original screen can be seen in Figure 5 below. Also, to make it simple and clear, we have also made a layout-only screen based on all AOI blocks.

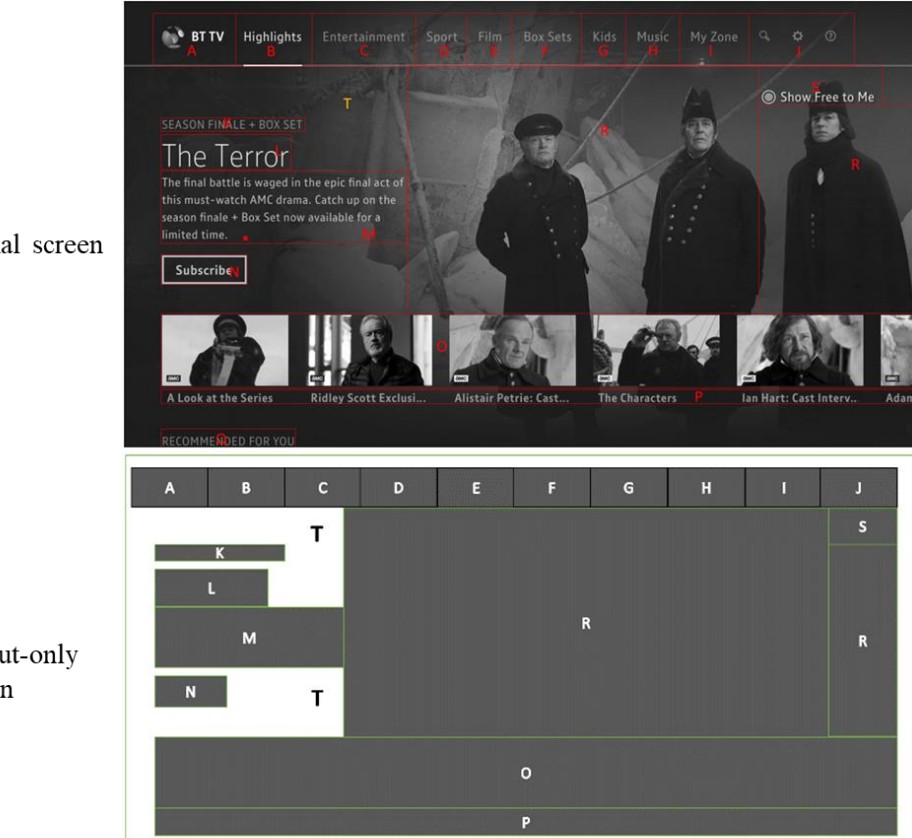

**Figure 5.** "Content viewing"–Original (**top**) & Layout-only (**bottom**) screens [16].

In Figure 5 above, block A has the BT TV logo, whereas blocks B through J include menu items. R features a large "hero" graphic for a programme being advertised. The K to N blocks provide additional information on this programme. S comprises shortcut buttons and overlaps R; hence, R is represented by two blocks. Both O and P display images and text as supplementary information. Q is positioned at the bottom of the page and provides a link to suggested shows. Block T refers to the area outside of the defined blocks, which is a non-AOI block.

The next step is to convert the gaze movement data into transitions between AOIs so that we can apply DTMCs. Unfortunately, although the Tobii Software can define an AOI and can provide statistics based on the metrics in relation to those AOIs, such as the mean time to first fixation on each AOI for the user, it does not provide a time-sequential list of the AOIs the user was fixating on for the duration of the recording, which is necessary for the Markov Model. However, the raw Tobii data, which provides X and Y gaze point co-ordinates of the eye, can be used to do this. The X and Y co-ordinates axis are shown in Figure 6 below, where the origin begins in the top left-hand corner; however, the X and Y axis both extend further in the horizontal and vertical directions than shown.

Therefore, based on a 1920 × 1080 resolution screen in our study, we have extracted the co-ordinates information for each AOI block for both scenarios. A sample of AOIs and co-ordinates of each corner can be seen in Figure 7 below. Each cell (e.g., (349,128)) under the columns shows the x and y axis coordinates, respectively. For the AOI column, E* and F* refer to the AOIs only available on the screen A.

Then a Java function was created to convert gaze point coordinates into the block (AOI) labels, as is shown in Figure 8 below.

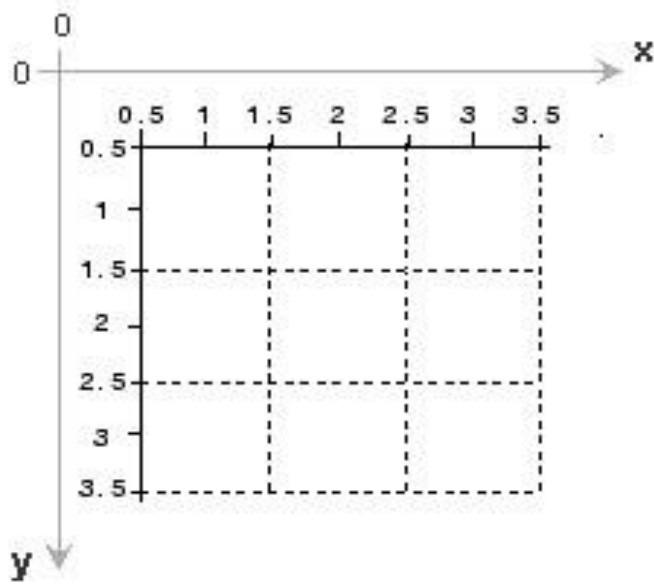

**Figure 6.** Tobii Studio Image Co-Ordinate System.

| AOI | Upper-Left Corner | Upper-Right Corner | Lower-Left Corner | Lower-Right Corner |
|---|---|---|---|---|
| A | (349,128) | (655,128) | (349,299) | (655,299) |
| B | (709,128) | (1379,128) | (709,299) | (1379,299) |
| C | (349,376) | (602,376) | (349,451) | (602,451) |
| D | (619,376) | (872,376) | (619,451) | (872,451) |
| E *(screen A only) | (349,527) | (602,527) | (349,602) | (602,602) |
| F *(screen A only) | (619,527) | (872,527) | (619,602) | (872,602) |
| Z (Non-AOI On screen) | | | | |

**Figure 7.** AOIs and corner co-ordinates–"content purchase" screens.

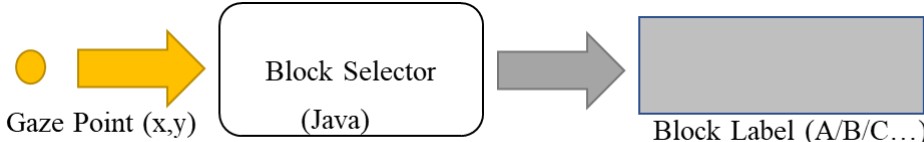

**Figure 8.** Conversion from Gaze Coordinate to Block Label [16].

The function results in an extra column named 'block_letter' in the MySQL table, which we created to store processed gaze movement data (see Figure 9). It indicates which AOI on the screen the current gaze movement falls in.

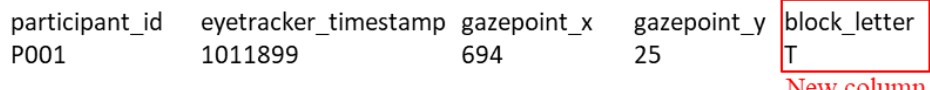

| participant_id | eyetracker_timestamp | gazepoint_x | gazepoint_y | block_letter |
|---|---|---|---|---|
| P001 | 1011899 | 694 | 25 | T |

New column

**Figure 9.** Example of gaze movement data in MySQL table.

In this way, we have a sequential list of AOIs from the original gaze movement data, which is suitable for a discrete-time Markov model. The full set of all possible states (also known as state space *S*) for both scenarios are as follows:

For **"content purchase"** screens:

- $S$ = {"A," "B," "C," "D," "E," "F," "Z"} – Screen A;
- $S$ = {"A," "B," "C," "D," "Z"} – Screen B;
- For **"content viewing"** screens:
- $S$ = {"A," "B," "C," "Dl," "E," … "T"}.

2.  Initial state probability distribution

For a Markov Model, if it exists, $X_0$ refers to the state when $t = 0$, and the initial probability distribution is defined by:

$$P(X_0 = s_i) = \frac{count(X_0 = s_i)}{count(X_0)} \tag{18}$$

The distribution is also called the initial state vector of the Markov chain (see Equation (7)). For instance, for the '**content viewing**' scenario, among all 14 participants, their eye gaze fell on different areas at first sight. The details are listed below:

- Two participants looked at AOI-E;
- One participant looked at AOI-I;
- Two participants looked at AOI-L;
- Two participants looked at AOI-M;
- One participant looked at AOI-O;
- Four participants looked at AOI-R;
- Two participants looked at AOI-T.

Therefore, the initial state distribution is estimated and displayed in Figure 10 as follows:

3.  Transition matrix

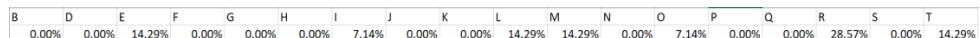

| B | D | E | F | G | H | I | J | K | L | M | N | O | P | Q | R | S | T |
|---|---|---|---|---|---|---|---|---|---|---|---|---|---|---|---|---|---|
| 0.00% | 0.00% | 14.29% | 0.00% | 0.00% | 0.00% | 7.14% | 0.00% | 0.00% | 14.29% | 14.29% | 0.00% | 7.14% | 0.00% | 0.00% | 28.57% | 0.00% | 14.29% |

**Figure 10.** Initial State Distribution–"content viewing" screen.

A transition matrix in the Markov model, in principle, is a reflection of how states change during the whole process. In our study, the transition matrix for a given state space infers how participants move their eyes on the screen; in other words, it illustrates the "jump" from the current state to the next state in the Markov Model created for each scenario.

In the MySQL database that we use to store eye tracking log data, after data pre-processing (AOI-categorization, non-relevant record filtering out, etc.), a sequence of states was determined for each participant, combined with the timestamp information and block labels. Furthermore, in the sequence, each pair of states indicates a one-step transition $(X_t, X_{t+1})$ in the DTMC model. In that regard, the state change pairs can be generated by simply applying an iteration of a length-2 sliding window over each sequence. In this paper, the function was implemented in Java. An example of the function is displayed in Figure 11 below:

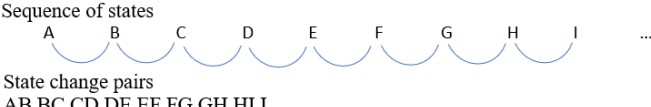

**Figure 11.** Example of state-change pair generation (from sequential events to transition pairs).

Subsequently, for both scenarios, transition matrices were defined by employing the maximum likelihood estimator (MLE) [17] of the $p_{ij}$ entry, where $n_{ij}$ stands for the number of pairs $(X_t = s_i, X_{t+1} = s_j)$ found in the whole dataset. The calculation for each $p_{ij}$ is presented by the equation below [17]:

$$\hat{p}_{ij}^{MLE} = \frac{n_{ij}}{\sum_{u=1}^{k} n_{iu}} \tag{19}$$

The standard errors of the $\hat{p}_{ij}^{MLE}$ values [17] were then generated by:

$$SE_{ij} = \frac{\hat{p}_{ij}^{MLE}}{\sqrt{n_{ij}}} \tag{20}$$

The transition matrix generation was implemented in Java by exploiting the features of hash maps and two-dimensional arrays. Subsequently, the transition matrices were exported into .csv files by the Java function and later were used in either MATLAB or R Studio to create DTMC models, implemented by the functions we have demonstrated in Section 3.4.2. For example, in the 'content viewing' scenario, R studio was used to read the .csv file of the created transition matrix. Using the "markovchain" package, a DTMC object was generated based on the transition matrix. We also acquired crucial Markov Chain properties, such as the classification of states. Next, the 'steadyStates()' R function was used to compute the steady-state vector (see Equation (7)). Using the function 'meanFirstPassageTime()', a matrix of mean first passage times between state $s_i$ and $s_j$ (let $s_i$ be the initial state) was also generated. Equation (13) was then used to construct the 'most likely trajectory,' which illustrated the course of the most likely state at each stage of the Markov process. In this instance, it depicts the most prominent way in which participants travel their eyes across the screen.

### 3.4.4. Summary of Data Pipeline

The pipeline of data throughout our study from collection to analysis is presented in Figure 12 below:

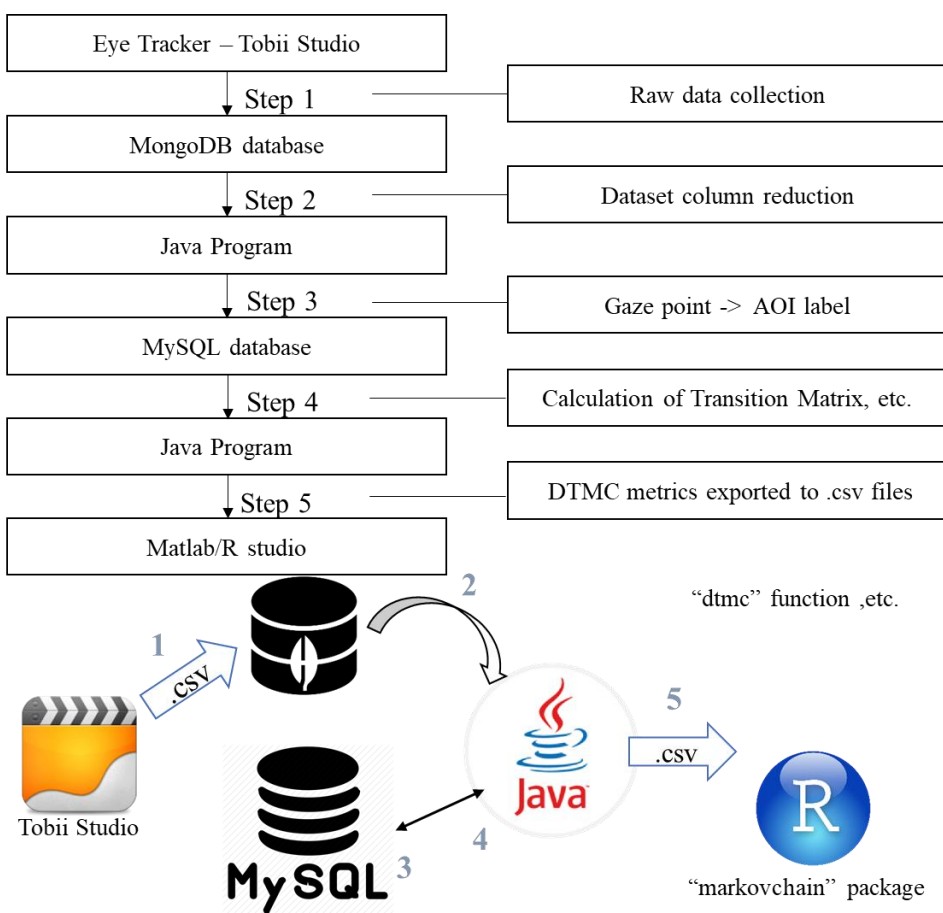

**Figure 12.** Data pipeline throughout our study [16].

We can see that the raw data originated from the eye tracker, was initially stored by Tobii Studio, and then exported via .csv files, thereby being transferred to a MongoDB database (Step 1). Afterwards, the data was cleansed using a Java program (Step 2) and then passed into a MySQL database (Step 3). Before records could be saved into the MySQL database, the Java program converted gaze point coordinates to block labels. In Step 4, a Java program accessed the MySQL database and calculated the properties of

the DTMC model (i.e., the transition matrix and the initial state distribution). Lastly, in Step 5, the Java program created some .csv files containing DTMC metrics for MATLAB (**"content purchase"** scenario) or R Studio (**"content viewing"** scenario) to employ. Then, in MATLAB, functions such as "dtmc" and "graphplot" were implemented, while in R Studio, we used functions mostly from the "markovchain" package, subsequently to create and observe DTMC models.

### 3.5. DTMC Visualisation

Once DTMC models were created, to have a better understanding of them so that we could make more sense of the gaze movement process, several attempts were made to visualize the model in different ways. One option is to use the built-in function in MATLAB, which is similar to using the R plotting function in R Studio. However, those functions generate static plots and can be overcrowded in the case of a model with many states. A more interactive plot, better to show the transition process as well, was therefore used for our study.

Therefore, in order to enhance comprehension and readability, the DTMC model given in this work would incorporate animations, illustrating a clear graph and modelling the transition process. To do this, the "D3.js" JavaScript package was used to generate an interactive display of the DTMC, complete with the animation of each state transition.

## 4. Results

This section may be divided into subheadings. It should provide a concise and precise description of the experimental results, their interpretation, as well as the experimental conclusions that can be drawn.

### 4.1. DTMC–"Content Purchase" Screens

In order to incorporate purchase flow through to the payment screen into the Markov Model, an extra "Payment" state has been included to model the trajectories of the BT Player customers' fixations, to better understand which AOIs they are fixating on before they go through to payment and which they are not. The "Payment" state will be both the last row and column in transition matrices.

Once the BT Player customer goes through to the payment screen, they are classified as choosing to pay for the TV Show/Film, and therefore if they choose to go back from payment to the previous screen, this is regarded as a failure to go through with payment. This means the payment state is accessible from any of the states on screen A or B, but the converse is not true, and the other states are not accessible from the payment state. Therefore, these transition matrices are no longer irreducible, and the steady state cannot be calculated.

Furthermore, the newly introduced "Payment" state is an absorbing state as once the Markov chain enters the state, it will remain there with a probability of one by definition of the state. Therefore, the expected number of steps to absorption and therefore expected time to absorption can be calculated for each state. In our case, it is possible when including all AOIs states and the non-AOI (Z) state as it is representative of the whole model. In a word, the Payment screen will simply be defined as an additional state in the Markov chain, "Payment". The inclusion of this state allows the expected time taken by the user to reach payment, based on the Markov Model.

#### 4.1.1. Transition Matrix–Screen A&B

The transition matrix for the initial states A-F for screen A, as well as including the non-AOI (Z) and "Payment" states, have been generated. This allows the expected number of steps and time to absorb into the "Payment" state to be calculated afterwards. The directed graph, which is the visualisation of the DTMC in MATLAB, has also been created. Figure 13a,b, respectively, show the transition matrix and plotting for the DTMC model of screen A.

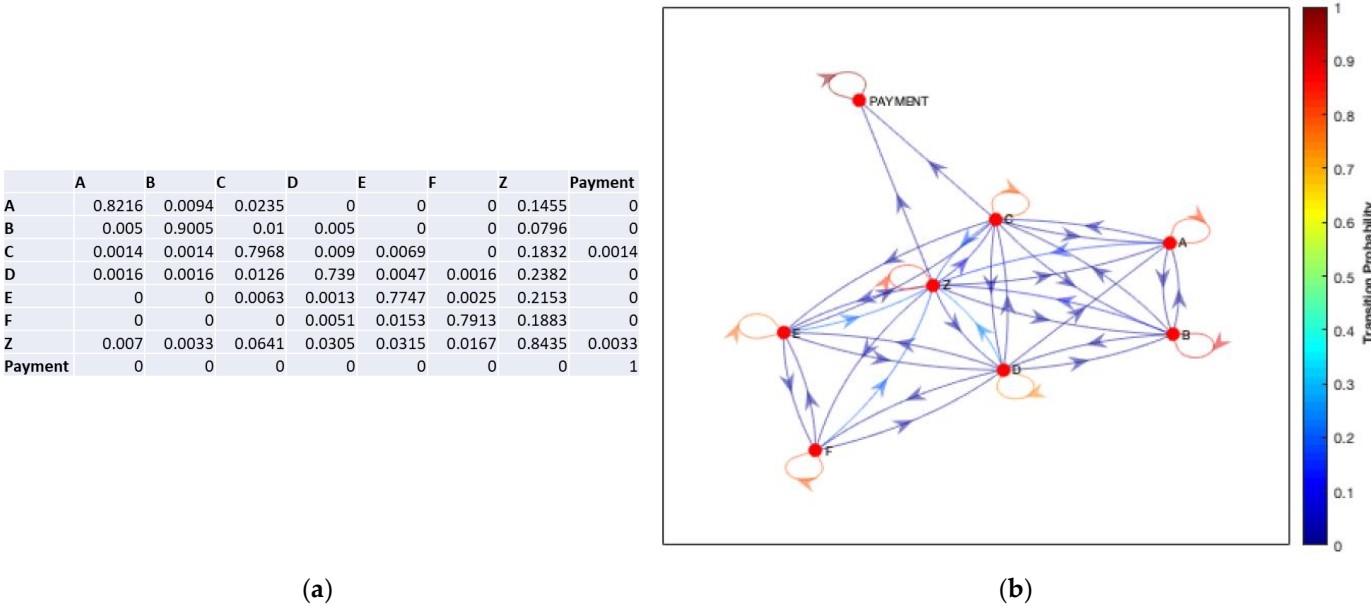

|         | A      | B      | C      | D      | E      | F      | Z      | Payment |
|---------|--------|--------|--------|--------|--------|--------|--------|---------|
| A       | 0.8216 | 0.0094 | 0.0235 | 0      | 0      | 0      | 0.1455 | 0       |
| B       | 0.005  | 0.9005 | 0.01   | 0.005  | 0      | 0      | 0.0796 | 0       |
| C       | 0.0014 | 0.0014 | 0.7968 | 0.009  | 0.0069 | 0      | 0.1832 | 0.0014  |
| D       | 0.0016 | 0.0016 | 0.0126 | 0.739  | 0.0047 | 0.0016 | 0.2382 | 0       |
| E       | 0      | 0      | 0.0063 | 0.0013 | 0.7747 | 0.0025 | 0.2153 | 0       |
| F       | 0      | 0      | 0      | 0.0051 | 0.0153 | 0.7913 | 0.1883 | 0       |
| Z       | 0.007  | 0.0033 | 0.0641 | 0.0305 | 0.0315 | 0.0167 | 0.8435 | 0.0033  |
| Payment | 0      | 0      | 0      | 0      | 0      | 0      | 0      | 1       |

(**a**)                                                                                          (**b**)

**Figure 13.** (**a**) Transition Matrix for Screen A; (**b**) DTMC Plot for Screen A.

In terms of the transition matrix in Figure 13a, the highest probabilities fall along the diagonal, suggesting that the most frequent state change is to remain within that state for the next period, which makes sense for this data as the timestamps are recorded every 00:00:00.165 s and if the user is focusing on an image or reading some text then this will require a greater period than jumping from fixating on one AOI to the next. In both Figure 13a,b, state A is accessible from state D, but this is not the case vice-versa, as state D is not directly accessible from state A. In terms of AOIs, this suggests that the viewer's eye trajectories can begin at "Buy to own SD" and return to the "Packshot image," but never from the "Packshot" to the "Buy to own SD" button. This would further suggest that viewers either glance directly to the right from the "Packshot" to the information or straight down to the "Buy to own HD" button. This could potentially be due to the "Buy to own HD" button being highlighted in a different colour than the rest of the buttons, therefore drawing the viewers' eyes immediately.

Similarly, the transition matrix and directed graph for screen B are shown in Figure 14a,b below.

It can be seen that the same pattern is emerging as with screen A, albeit with fewer states available. It is interesting that despite being a different screen, the users never directly access from state A to D, only D to A, suggesting the "Buy SD" option does not immediately draw their interest from the "Packshot", and instead, the user fixates on the information or the "Buy HD" option after looking at the Packshot image. This agrees with the concept that we observe computer screens horizontally and vertically, in a traditional "F" shape, much more frequently than diagonally. Once again, in the transition matrix for Screen B in Figure 14a, the highest transitional probabilities are along the diagonal, where the fixations remain within the same state for a prolonged period.

For both screens, the non-AOI (Z) state communicates with all states as it covers a large area surrounding all of the AOIs. For screen A, the "Payment" state can be accessed from both state C and the non-AOI state, while for screen B, the "Payment" state is accessible from states C and D and the non-AOI state(Z). This could be a result of users reading the "Buy or Rent?" information at the bottom of the page before proceeding to payment. The bottom row of each matrix shows the Payment state is absorbing, with all of the probabilities of the payment state reaching a different state equal to 0, while the probability of remaining within the payment state once it has been entered is equal to 1. As this is an absorbing state and as Figures 13 and 14 represent an accurate model of each screen, now the expected time to absorption (Payment) can be calculated.

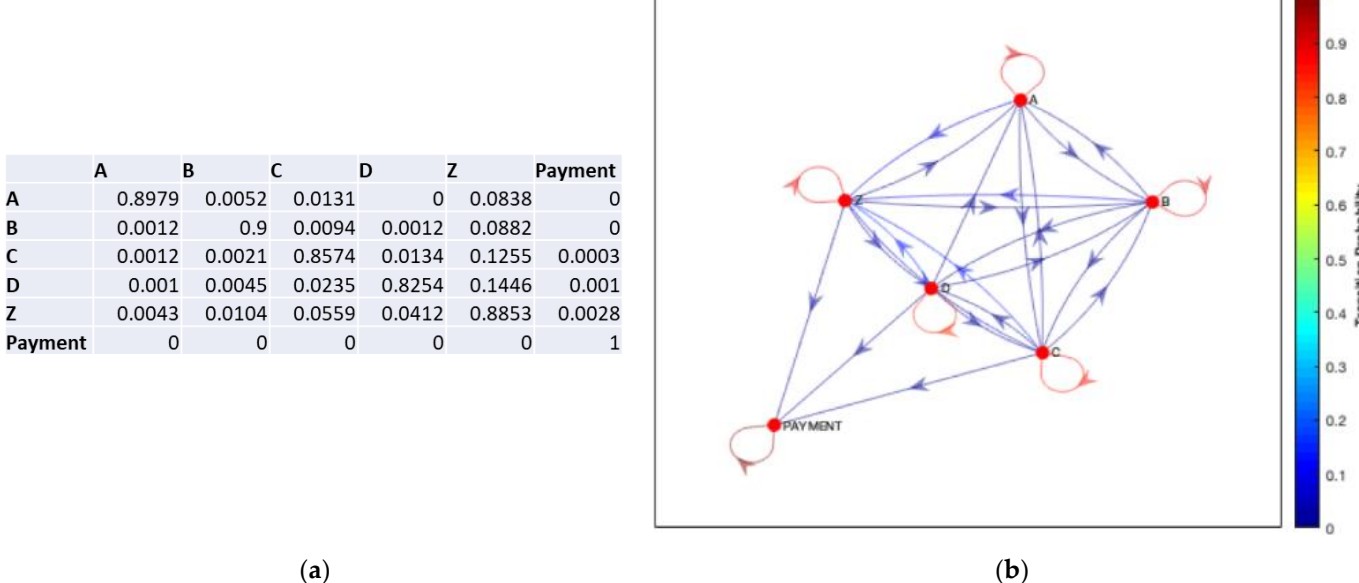

| | A | B | C | D | Z | Payment |
|---|---|---|---|---|---|---|
| **A** | 0.8979 | 0.0052 | 0.0131 | 0 | 0.0838 | 0 |
| **B** | 0.0012 | 0.9 | 0.0094 | 0.0012 | 0.0882 | 0 |
| **C** | 0.0012 | 0.0021 | 0.8574 | 0.0134 | 0.1255 | 0.0003 |
| **D** | 0.001 | 0.0045 | 0.0235 | 0.8254 | 0.1446 | 0.001 |
| **Z** | 0.0043 | 0.0104 | 0.0559 | 0.0412 | 0.8853 | 0.0028 |
| **Payment** | 0 | 0 | 0 | 0 | 0 | 1 |

(**a**)

(**b**)

**Figure 14.** (**a**) Transition Matrix for Screen B; (**b**) DTMC Plot for Screen B.

From the perspective of HCI analysis, for both screens A and B, state C is the 'Buy to Own HD button', which is on the left-hand side and highlighted in a pale grey colour which heavily contrasts against the dark grey background, while the other buttons appear in a much darker purple colour that is less of a contrast against the black background. As the button is on the left-hand side, the user's eyes will naturally be drawn more to it than the buttons on the right-hand side. Also, from the Gestalt principles [33–35], the brain will naturally group the purple buttons (states D–F on screen A), so the eye will naturally be drawn to the contrast button. As this is the most expensive option, this is the ideal position and colour for this button to drive sales. In addition, the high probabilities of entering the non-AOI(Z) state could be interpreted as users reading the information at the bottom of the page relating to the 'Buy' or 'Buy or Rent?' possibilities, or they could be fixating on the cancel button if they have decided they no longer want to continue with the purchase. Another reason for this could be due to the fact that users are on a different screen before proceeding to the purchase content screen (A or B), and therefore they could be fixating on a different area of the screen that is then a non-AOI area of screen A or B. If this is the case, the user will continue to fixate on that area initially for a short period before then focusing on a different area of the new screen.

4.1.2. Expected Time to Payment–Screen A&B

For screen A, the transition matrix can be taken and separated into subsections, as shown in Equation (15). Given there are 7 transient states and 1 absorbing state, $m = 7$ and $n = 1$, therefore we calculate the $Q$ and $R$ sections in the transition matrix, and the results are shown in Figure 15 below.

$$Q = \begin{bmatrix} 0.8216 & 0.0094 & 0.0235 & 0 & 0 & 0 & 0.1455 \\ 0.005 & 0.9005 & 0.01 & 0.005 & 0 & 0 & 0.0796 \\ 0.0014 & 0.0014 & 0.7968 & 0.009 & 0.0069 & 0 & 0.1832 \\ 0.0016 & 0.0016 & 0.0126 & 0.739 & 0.0047 & 0.0016 & 0.2382 \\ 0 & 0 & 0.0063 & 0.0013 & 0.7747 & 0.0025 & 0.2153 \\ 0 & 0 & 0 & 0.0051 & 0.0153 & 0.7913 & 0.1883 \\ 0.007 & 0.0033 & 0.0641 & 0.0305 & 0.0315 & 0.0167 & 0.8435 \end{bmatrix}$$

$$R = \begin{bmatrix} 0 \\ 0 \\ 0.0014 \\ 0 \\ 0 \\ 0 \\ 0.0033 \end{bmatrix}$$

(**a**)

(**b**)

**Figure 15.** Screen A: (**a**) Section $Q$ in the transition matrix; (**b**) Section $R$ in the transition matrix.

The Fundamental Matrix can then be calculated using Equation 16, where $I_n$, in this case, will be an identity matrix of size $m = 7$, so we have $N$ in Figure 16 below.

$$N = \begin{bmatrix} 17.224 & 12.114 & 88.611 & 34.607 & 41.515 & 21.713 & 261.81 \\ 11.899 & 21.674 & 88.574 & 34.825 & 41.562 & 21.739 & 262.11 \\ 11.575 & 11.584 & 92.386 & 34.553 & 41.417 & 21.594 & 260.35 \\ 11.642 & 11.647 & 88.226 & 38.443 & 41.605 & 21.754 & 261.95 \\ 11.612 & 11.587 & 88.167 & 34.638 & 45.986 & 21.789 & 262.1 \\ 11.608 & 11.584 & 88.021 & 34.698 & 41.857 & 26.527 & 262.03 \\ 11.607 & 11.582 & 88.004 & 34.601 & 41.528 & 21.73 & 262.03 \end{bmatrix}$$

**Figure 16.** Fundamental Matrix $N$ for DTMC–Screen A.

Next, using Equation 17, the expected number of steps to absorption can be calculated, as given there are 7 transient states, $c$ will be a $7 \times 1$ column vector of 1's, and therefore $t$ will also be a matrix of size $7 \times 1$. Besides, given that $t$ represents the expected number of steps to absorption for all states, then $t_i$ represents the expected number of steps to absorption from a given state $i$. In this case, the states are A, B, C, D, E, F and Z. Also, considering the created discrete Markov chain follows a time step of 0.165 s between states, the expected number of steps for each state will be multiplied by 0.165 to get the real expected time to absorption from each state. Table 2 shows the calculated expected time to absorption from each state.

**Table 2.** Expected time to absorption from each state–DTMC for screen A.

| AOI/State Name | Expected Time to Absorption |
| --- | --- |
| A | 7.88 s |
| B | 7.96 s |
| C | 7.81 s |
| D | 7.84 s |
| E | 7.85 s |
| F | 7.86 s |
| Z | 7.77 s |

From these calculations, it can be inferred that despite the initial state (or AOI on screen A) being fixated on, it takes relatively the same time to reach the payment state, with less than 0.2 s of a difference between the 'quickest' initial state Z and the 'slowest' initial state B.

Likewise, the calculations by Equations (15)–(17) are applied to screen B. For this screen, there are 5 transient states and 1 absorbing state, so $m = 5$ and $n = 1$. The Q and R sections in the transition matrix and the results are shown in Figure 17 below.

$$Q = \begin{bmatrix} 0.8979 & 0.0052 & 0.0131 & 0 & 0.0838 \\ 0.0012 & 0.9 & 0.0094 & 0.0012 & 0.0882 \\ 0.0012 & 0.0021 & 0.8574 & 0.0134 & 0.1256 \\ 0.001 & 0.0045 & 0.0235 & 0.8254 & 0.1446 \\ 0.0043 & 0.0104 & 0.0559 & 0.0412 & 0.8853 \end{bmatrix} \qquad R = \begin{bmatrix} 0 \\ 0 \\ 0.0003 \\ 0.0012 \\ 1 \end{bmatrix}$$

(**a**)         (**b**)

**Figure 17.** Screen B: (**a**) Section $Q$ in the transition matrix; (**b**) Section $R$ in the transition matrix.

The Fundamental matrix, in this case, by Equation (16), is shown in Figure 18.

$$N = \begin{bmatrix} 25.404 & 39.336 & 136.98 & 82.125 & 302.34 \\ 15.719 & 48.831 & 136.73 & 82.173 & 302.35 \\ 15.647 & 38.892 & 42.84 & 82.406 & 301.64 \\ 15.574 & 38.866 & 136.23 & 87.382 & 300.6 \\ 15.597 & 38.817 & 136.08 & 82.078 & 302.45 \end{bmatrix}$$

**Figure 18.** Fundamental Matrix $N$ for DTMC–Screen B.

Again, using Equation (17), the expected number of steps to absorption can be calculated, as given there are 5 transient states, $c$ will be a $5 \times 1$ column vector of 1's, and therefore $t$ will also be a matrix of size $5 \times 1$. For screen B, the states are A, B, C, D, and Z. Table 3 below shows the calculated expected time to absorption for the DTMC on screen B.

**Table 3.** Expected time to absorption from each state–DTMC for screen B.

| AOI/State Name | Expected Time to Absorption |
|---|---|
| A | 9.67 s |
| B | 9.67 s |
| C | 9.59 s |
| D | 9.54 s |
| Z | 9.49 s |

According to Table 3, it can be concluded that irrespective of the initial state, it takes approximately the same expected time to the absorption state (Payment screen), with almost the same range between the 'quickest', again the Z state, and the 'slowest', both A and B have the same expected time.

It is also interesting to note that although there are more AOIs on screen A (6: A–F) than on screen B (4: A–D), the expected time to reach the payment screen on screen A is almost 2 s faster than B. Despite having additional options to choose from, the TV customer is more efficient in making their choice when there are "Buy and Rent" HD and SD options rather than when there are only "Buy" options available.

*4.2. DTMC–"Content Viewing" Screens*

After developing the DTMC model, it was revealed that given the restricted time allotted for this particular activity, participants did not focus on certain blocks as they did in previous tasks. Therefore, the state space of the DTMC model is finalised as $S = \{B, D, E, F, G, H, I, J, K, L, M, N, O, P, Q, R, S, T\}$. The Markov Modelling of the **"content viewing"** screen is implemented in R Studio by the "markovchain" package and R functions belonging to it, which was previously discussed in Section 3.4.2.

In R studio, after the DTMC object is created, the properties of the model can be checked by calling the object itself as well as specific functions, see Figure 19. In the example, the created DTMC object is named "dtmcA", and the R function "communicateingClasses()" is executed.

```
MarkovChain A  Markov chain that is composed by:
Closed classes:
B D E F G H I J K L M N O P Q R S T
Recurrent classes:
{B,D,E,F,G,H,I,J,K,L,M,N,O,P,Q,R,S,T}
Transient classes:
NONE
The Markov chain is irreducible
The absorbing states are: NONE
> communicatingClasses(dtmcA)
[[1]]
 [1] "B" "D" "E" "F" "G" "H" "I" "J" "K" "L" "M" "N" "O" "P" "Q" "R" "S" "T"
```

**Figure 19.** Markov Model properties–"content viewing" screen.

Observe that there is only one communicating class in the model and that all states communicate. When a participant looks at one block on the page, there is always a path in the subsequent n steps of the gaze movement by which they will finally look at the remaining blocks in the DTMC model. In addition, it strengthens the likelihood that participants explored the page across all AOIs throughout the trial. There is no absorbing state, i.e., participants are unlikely to concentrate on a single AOI for the duration of the experiment.

The transition matrix of the DTMC, as well as the matrix of mean first passage times (see Equation (14)), are presented in the 18 × 18 matrices in Figures 20 and 21 below.

|   | B | D | E | F | G | H | I | J | K | L | M | N | O | P | Q | R | S | T |
|---|---|---|---|---|---|---|---|---|---|---|---|---|---|---|---|---|---|---|
| **B** | 0.918 | 0.000 | 0.000 | 0.000 | 0.000 | 0.000 | 0.000 | 0.000 | 0.000 | 0.000 | 0.000 | 0.000 | 0.000 | 0.000 | 0.000 | 0.000 | 0.000 | 0.082 |
| **D** | 0.000 | 0.000 | 0.333 | 0.000 | 0.000 | 0.000 | 0.000 | 0.000 | 0.000 | 0.000 | 0.000 | 0.000 | 0.000 | 0.000 | 0.000 | 0.667 | 0.000 | 0.000 |
| **E** | 0.000 | 0.000 | 0.788 | 0.030 | 0.000 | 0.000 | 0.000 | 0.000 | 0.000 | 0.000 | 0.000 | 0.000 | 0.000 | 0.000 | 0.000 | 0.136 | 0.000 | 0.045 |
| **F** | 0.000 | 0.000 | 0.031 | 0.875 | 0.000 | 0.000 | 0.000 | 0.000 | 0.000 | 0.000 | 0.031 | 0.000 | 0.000 | 0.000 | 0.000 | 0.031 | 0.000 | 0.031 |
| **G** | 0.000 | 0.000 | 0.000 | 0.000 | 0.909 | 0.000 | 0.000 | 0.000 | 0.000 | 0.000 | 0.000 | 0.000 | 0.000 | 0.000 | 0.000 | 0.045 | 0.000 | 0.045 |
| **H** | 0.000 | 0.000 | 0.000 | 0.000 | 0.000 | 0.833 | 0.000 | 0.000 | 0.000 | 0.000 | 0.000 | 0.000 | 0.000 | 0.000 | 0.000 | 0.167 | 0.000 | 0.000 |
| **I** | 0.000 | 0.000 | 0.000 | 0.000 | 0.000 | 0.000 | 0.897 | 0.034 | 0.000 | 0.000 | 0.000 | 0.000 | 0.000 | 0.000 | 0.000 | 0.034 | 0.000 | 0.034 |
| **J** | 0.000 | 0.000 | 0.000 | 0.000 | 0.000 | 0.000 | 0.000 | 0.800 | 0.000 | 0.000 | 0.000 | 0.000 | 0.000 | 0.000 | 0.000 | 0.143 | 0.000 | 0.057 |
| **K** | 0.000 | 0.000 | 0.000 | 0.000 | 0.000 | 0.000 | 0.000 | 0.000 | 0.543 | 0.086 | 0.057 | 0.000 | 0.000 | 0.000 | 0.000 | 0.000 | 0.000 | 0.314 |
| **L** | 0.000 | 0.000 | 0.000 | 0.000 | 0.000 | 0.000 | 0.000 | 0.000 | 0.014 | 0.839 | 0.070 | 0.007 | 0.000 | 0.007 | 0.000 | 0.007 | 0.000 | 0.056 |
| **M** | 0.000 | 0.000 | 0.000 | 0.000 | 0.000 | 0.000 | 0.000 | 0.000 | 0.007 | 0.030 | 0.852 | 0.026 | 0.016 | 0.000 | 0.000 | 0.016 | 0.000 | 0.053 |
| **N** | 0.000 | 0.000 | 0.000 | 0.000 | 0.000 | 0.000 | 0.000 | 0.000 | 0.000 | 0.000 | 0.027 | 0.770 | 0.005 | 0.000 | 0.000 | 0.000 | 0.000 | 0.197 |
| **O** | 0.000 | 0.000 | 0.000 | 0.000 | 0.000 | 0.000 | 0.000 | 0.000 | 0.000 | 0.000 | 0.001 | 0.001 | 0.890 | 0.046 | 0.001 | 0.004 | 0.000 | 0.057 |
| **P** | 0.000 | 0.000 | 0.000 | 0.000 | 0.000 | 0.000 | 0.000 | 0.000 | 0.000 | 0.005 | 0.000 | 0.000 | 0.236 | 0.482 | 0.000 | 0.000 | 0.000 | 0.277 |
| **Q** | 0.000 | 0.000 | 0.000 | 0.000 | 0.000 | 0.000 | 0.000 | 0.000 | 0.000 | 0.000 | 0.000 | 0.000 | 0.000 | 0.000 | 0.476 | 0.000 | 0.000 | 0.524 |
| **R** | 0.000 | 0.003 | 0.028 | 0.000 | 0.003 | 0.003 | 0.006 | 0.014 | 0.000 | 0.003 | 0.003 | 0.000 | 0.022 | 0.000 | 0.000 | 0.884 | 0.008 | 0.025 |
| **S** | 0.000 | 0.000 | 0.000 | 0.000 | 0.000 | 0.000 | 0.000 | 0.000 | 0.000 | 0.000 | 0.000 | 0.000 | 0.000 | 0.000 | 0.000 | 0.019 | 0.981 | 0.000 |
| **T** | 0.004 | 0.003 | 0.003 | 0.003 | 0.001 | 0.000 | 0.000 | 0.001 | 0.011 | 0.006 | 0.022 | 0.030 | 0.057 | 0.058 | 0.009 | 0.006 | 0.000 | 0.788 |

**Figure 20.** Transition matrix–"content viewing" screen [16].

|   | B | D | E | F | G | H | I | J | K | L | M | N | O | P | Q | R | S | T |
|---|---|---|---|---|---|---|---|---|---|---|---|---|---|---|---|---|---|---|
| **B** | 0.00 | 989.93 | 330.63 | 780.54 | 2010.46 | 4270.85 | 2143.06 | 698.24 | 258.25 | 220.04 | 99.36 | 100.25 | 36.15 | 44.59 | 356.72 | 150.63 | 1473.50 | 12.25 |
| **D** | 975.55 | 0.00 | 158.70 | 720.99 | 1956.35 | 4136.82 | 2009.03 | 588.84 | 275.84 | 228.15 | 110.57 | 118.06 | 44.52 | 59.10 | 375.57 | 16.60 | 1339.47 | 32.18 |
| **E** | 972.67 | 978.45 | 0.00 | 653.55 | 1969.37 | 4167.01 | 2039.22 | 613.74 | 272.97 | 226.50 | 106.76 | 115.09 | 43.98 | 57.31 | 372.90 | 46.79 | 1369.66 | 29.30 |
| **F** | 970.88 | 986.56 | 226.41 | 0.00 | 1987.75 | 4207.17 | 2079.38 | 647.19 | 267.85 | 217.14 | 84.21 | 109.40 | 44.19 | 56.39 | 371.30 | 86.95 | 1409.82 | 27.51 |
| **G** | 970.43 | 984.63 | 288.46 | 771.75 | 0.00 | 4200.41 | 2072.62 | 641.44 | 271.88 | 228.63 | 110.05 | 114.02 | 44.60 | 56.42 | 370.94 | 80.19 | 1403.06 | 27.06 |
| **H** | 981.49 | 975.58 | 242.55 | 759.21 | 1954.34 | 0.00 | 1998.43 | 580.89 | 281.77 | 233.48 | 116.98 | 124.04 | 49.29 | 64.50 | 381.40 | 6.00 | 1328.87 | 38.13 |
| **I** | 973.06 | 984.38 | 282.95 | 771.01 | 1981.04 | 4190.86 | 0.00 | 429.96 | 274.35 | 230.37 | 112.08 | 116.50 | 46.32 | 58.62 | 373.48 | 70.64 | 1393.51 | 29.69 |
| **J** | 971.32 | 976.89 | 264.93 | 762.52 | 1967.59 | 4164.76 | 2036.97 | 0.00 | 272.27 | 226.85 | 109.16 | 114.46 | 42.75 | 56.03 | 371.57 | 44.54 | 1367.41 | 27.95 |
| **K** | 950.65 | 983.41 | 321.27 | 773.76 | 2002.33 | 4259.30 | 2131.52 | 687.75 | 0.00 | 166.90 | 72.44 | 90.25 | 29.48 | 38.49 | 351.60 | 139.08 | 1461.96 | 7.29 |
| **L** | 959.49 | 989.35 | 321.90 | 779.19 | 2005.26 | 4255.86 | 2128.07 | 686.27 | 232.07 | 0.00 | 55.28 | 90.70 | 35.49 | 44.76 | 360.17 | 135.64 | 1458.51 | 16.12 |
| **M** | 959.97 | 988.53 | 318.70 | 778.15 | 2003.10 | 4250.84 | 2123.06 | 682.13 | 244.64 | 174.80 | 0.00 | 84.49 | 33.60 | 45.40 | 360.43 | 130.62 | 1453.49 | 16.60 |
| **N** | 950.03 | 983.62 | 322.99 | 774.11 | 2003.39 | 4262.19 | 2134.40 | 690.07 | 250.51 | 208.50 | 81.39 | 0.00 | 28.85 | 38.17 | 350.98 | 141.97 | 1464.84 | 6.66 |
| **O** | 957.27 | 989.88 | 327.45 | 780.19 | 2008.63 | 4265.27 | 2137.48 | 693.82 | 259.46 | 219.61 | 99.29 | 100.60 | 0.00 | 28.79 | 354.83 | 145.05 | 1467.92 | 13.90 |
| **P** | 951.78 | 985.27 | 324.48 | 775.75 | 2004.95 | 4263.55 | 2135.76 | 691.49 | 253.95 | 213.29 | 94.32 | 95.70 | 15.03 | 0.00 | 351.27 | 143.33 | 1466.20 | 8.41 |
| **Q** | 945.28 | 979.59 | 320.29 | 770.20 | 2000.12 | 4260.51 | 2132.72 | 687.90 | 247.91 | 209.70 | 89.02 | 89.91 | 25.81 | 34.25 | 0.00 | 140.29 | 1463.16 | 1.91 |
| **R** | 975.49 | 969.58 | 236.55 | 753.21 | 1948.34 | 4120.22 | 1992.43 | 574.89 | 275.77 | 227.48 | 110.98 | 118.04 | 43.29 | 58.50 | 375.40 | 0.00 | 1322.87 | 32.13 |
| **S** | 1027.99 | 1022.08 | 289.05 | 805.71 | 2000.84 | 4172.72 | 2044.93 | 627.39 | 328.27 | 279.98 | 163.48 | 170.54 | 95.79 | 111.00 | 427.90 | 52.50 | 0.00 | 84.63 |
| **T** | 943.37 | 977.68 | 318.38 | 768.29 | 1998.21 | 4258.60 | 2130.81 | 685.99 | 246.00 | 207.79 | 87.11 | 88.00 | 23.90 | 32.34 | 344.47 | 138.38 | 1461.25 | 0.00 |

**Figure 21.** Matrix of mean first passage time–"content viewing" screen [16].

In the row of R of the transition matrix (Figure 20), besides the diagonal cell (R,R), all other values in the cell are small. On the basis of this data, we may conclude that when participants notice AOI R (background image), they tend to study it further rather than moving on to other places. In addition, columns R and T have the greatest number of values, indicating a high chance of transition from other states in the Markov Model, indicating

that participants are likely to shift their attention to AOI-R or AOI-T from other AOIs. It indicates that the backdrop image is appealing to participants for AOI-R. T represents non-AOI portions of the display; it is a segment between many blocks. Participants shift their eyes to AOI-T, presumably because they will observe another AOI via T.

In Figure 21, the value in each cell is related to the number of steps before the DTMC reaches the destination state from the initial state. The diagonal values are 0 s because it takes no step to reach the state from itself. The values can also reflect the actual time (time = step number × eye tracker recording interval) participants spend to move their gaze from one block to another. On the other hand, the values also reflect the connections between every two blocks regarding the page layout. For instance, from the initial state H, it takes the chain Six steps to reach state R (cell(H, R)). Six is the minimum value among all other cases at which the chain reaches state R from the other initial states. The reason is that, according to the page layout (Figure 5), block H is adjacent to block R. Moreover, cell (H,R) has the lowest value in the matrix, indicating that once participants see the music button, they will quickly gaze at the backdrop image. In addition, column H has the highest average value compared to the other columns, indicating that it takes the longest time for the chain to reach state H from other states. Thus, the matrix of the "content viewing" page (Figure 21) reveals that participants are likely not as interested in the music button since they opt to view other sections first and position the music button last.

Apart from the matrices, Table 4 below shows the probability distribution over initial states and steady states for the DTMC model.

**Table 4.** Initial state and steady-state distribution–DTMC for "content viewing" screen.

| State | Initial Probability | Steady Probability |
|---|---|---|
| B | 0 | 0.013 |
| D | 0 | 0.001 |
| E | 0.143 | 0.018 |
| F | 0 | 0.010 |
| G | 0 | 0.006 |
| H | 0 | 0.002 |
| I | 0.071 | 0.005 |
| J | 0 | 0.008 |
| K | 0 | 0.009 |
| L | 0.143 | 0.032 |
| M | 0.143 | 0.075 |
| N | 0 | 0.047 |
| O | 0.071 | 0.303 |
| P | 0 | 0.059 |
| Q | 0 | 0.005 |
| R | 0.286 | 0.088 |
| S | 0 | 0.038 |
| T | 0.143 | 0.281 |

Under the column of initial state probability, State R has the highest value, indicating that all participants are more likely to look at state R first when they reach this 'content viewing' screen in BT Player, which can also be interpreted as "the background image attracts the most attention at first sight." The column of steady state probability reflects the likelihood that participants will view each block over time. State O, for example, has the highest value, indicating that the gaze is most likely to reach and terminate in that block ("programme packshots"). However, state D has the lowest score, indicating that after participants have explored the page for some time, they are least likely to view the 'sport button'.

Additionally, the "most likely trajectory" of the DTMC is generated by Equation (13), $S^{max} = \{R, \ldots, T, \ldots, O \ldots \}$, which is calculated by the simulation of Markov transitions in R. The trajectory is depicted and mapped to the real-world screen layout as the most likely gaze path on the screen, which is shown in Figure 22.

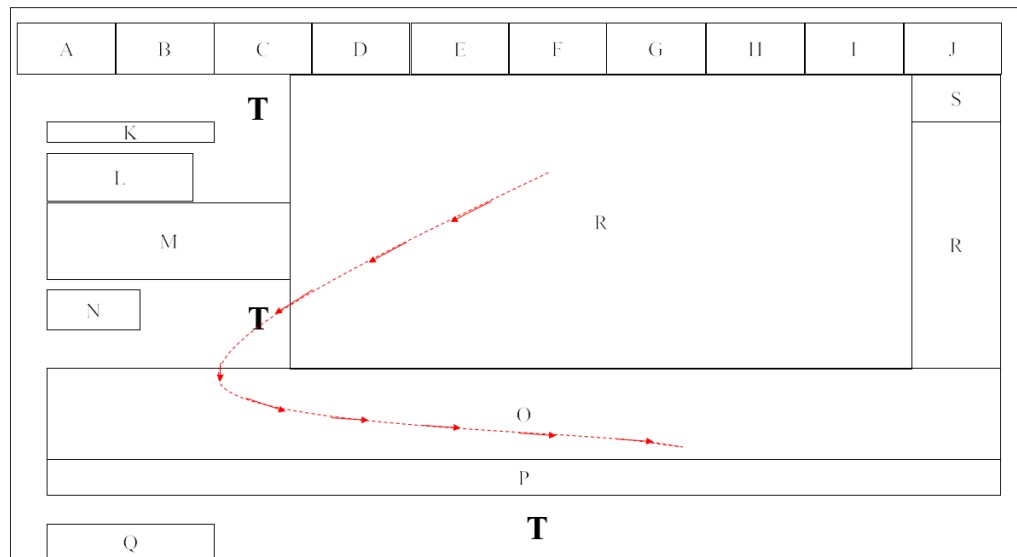

**Figure 22.** Most likely gaze trajectory–"content viewing" screen [16].

In addition, we have two implementations of the DTMC visualization in R and D3.js, as shown in Figures 23a and 23b, respectively.

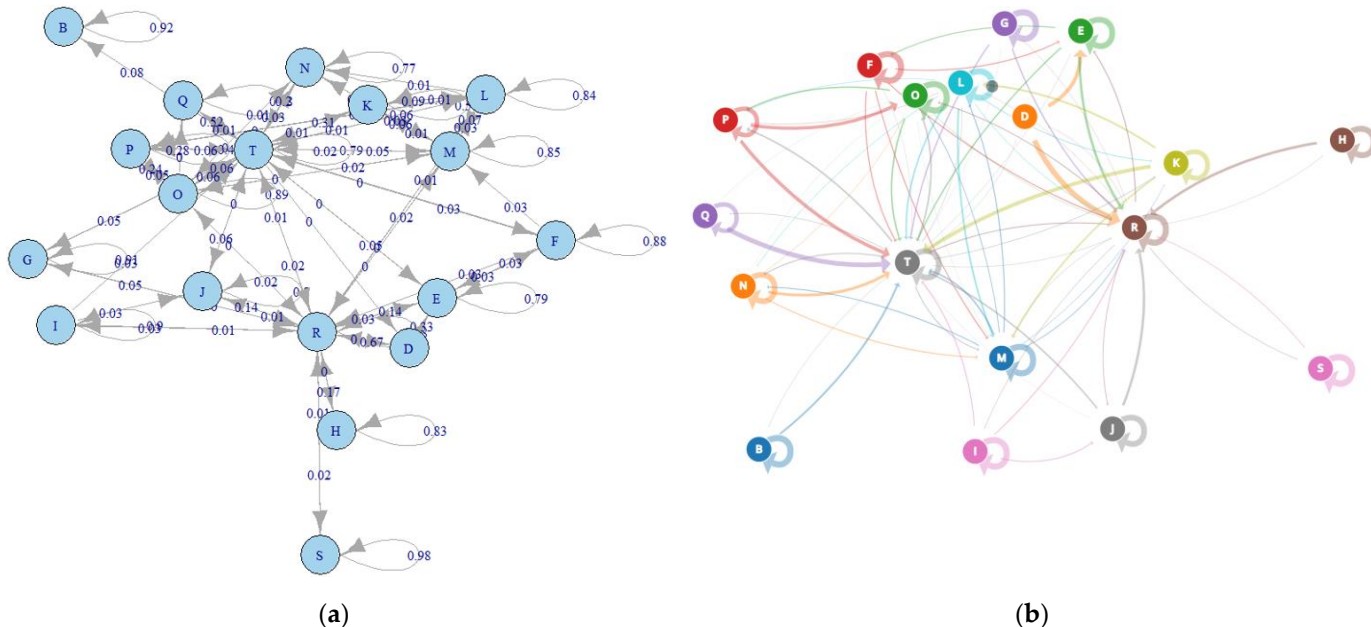

(**a**)                                                    (**b**)

**Figure 23.** "Content viewing" screen–DTMC visualisation: (**a**) R plotting; (**b**) D3.js plotting [16].

Intuitively, T is the state that most other states will jump to. Figure 23a is a static image, while Figure 23b is a screenshot of the animation of a Markov Model process on the browser. In Figure 23b, the thickness of arrows and directed lines indicates the value of probability. The animation of each transition is based on the transition matrix.

## 5. Discussion

Throughout the eye-tracking experiment, there were elements that were successful and went well, while there were also areas that were difficult and proved more challenging to complete. In other words, working with data involves a significant amount of understanding and then cleaning, or manipulation before the actual analysis can begin. This involved time to both understand the data and find a method that could be used to model it. For instance, a good level of prior knowledge about the raw data (e.g., gaze-point

coordinates and timestamps) from Tobii Studio was required for the analysis tasks. Meanwhile, the format of raw data brought some difficulties before the data modelling stage. Different requirements and tasks in relation to the experiment were also provided along with the statistical analysis based on these, and it was only after understanding the nature of the data that it became clear that gaze point co-ordinates in the raw Tobii data would need to be converted into AOI regions. As discussed in Section 3.4, a Java program was developed for:

- Capturing coordinates of AOI regions on the screen;
- Converting gaze point to AOI block letters;
- Raw data cleaning and transferring from Mongo DB to MySQL Workbench.

After a significant amount of research into data-mining algorithms such as Apriori and other methods, Markov-Chain Models were shown to be an effective technique to analyse this discrete time-sequential data as the raw data is recorded every 00:00:00.165 s, making it appropriate for the Discrete Time Markov Chain (DTMC) model.

MATLAB and R Studio provide powerful functions for Markov modelling, from simply plotting a Markov model with a directed graph to a greater magnitude of information available for Markov models, such as first passage time, mean time to recurrence, absorption and the extent of the statistical knowledge it would take to understand all of these terms and how to apply them. They have been used effectively to calculate the steady states of different matrices and the expected time to absorption for the Payment state; however, in hindsight, it would have been beneficial to better understand these complexities at the outset, which may have allowed the computation of each of these aspects to run more smoothly, rather than having to research more each time. The workflow of Markov modelling allowed for easy adaptation once the first screen transition matrix had been completed, as the same process just needed to be carried out on another screen and then implemented in the same manner on MATLAB/R Studio. This was beneficial as it was quite an efficient process, allowing for changes to be made if necessary.

The study was overall successful, and a lot was learned with regard to working with eye gaze movement data, exploring in-depth information for BT Player and mining the process of how participants interact with the UI on the screen. Specifically, the analysis of eye tracking data has enabled the two key research questions to be answered:

4. **"Content purchase"** scenario: when purchasing content (TV on Demand), what draws the eye? Is it the price, is it the quality, or is it something else?

It can be said that the price is of interest to the user; however, they are particularly focused on the "Buy to Own HD" option, potentially due to the fact that it is highlighted in a different background colour than the other price options. As it is also on the left-hand side of the page, it can be said that the user is more likely to fixate on the HD than the SD buttons, which are on the right-hand side. Taking into account several results from other aspects of the eye tracking experiment which are not covered in this study, a more general recommendation for Purchase Flow could be "Put important information in the top position of a screen to help the reading flow".

5. "**Content viewing"** scenario: when a Content Discovery page first loads, what are customers viewing? Are they drawn to the hero carousel, the navigation, or something else?

One outcome of the study was that large images first caught the attention of most people after loading a screen. Therefore, a recommendation could be made to use a large image to enhance a significant part and avoid large images in less important areas. These recommendations have helped to improve the design of the BT Player, as it could be confirmed that certain elements may need to be changed while others could remain the same in order to engage the customer and increase sales.

Furthermore, the research findings were shared with the BT product team and other individuals with domain knowledge of BT IPTV and TV customer behaviour. The DTMC-derived gaze movement trajectory, as well as other Markov properties such as steady-state

distribution, expected time to absorption, and so on, were found to be quite reasonable for the use cases. The product team provided some validation of the results by comparing them to the original layout design. Besides, our interactions with the BT team during the course of the eye-tracking study led to the following observations of more general business benefits, with feedback from BT as follows:

- Eye tracking studies can provide valuable inputs to a human-centred design approach for TV applications;
- Eye tracking results can show the order in which people focus on different parts of a TV application page, which enables designers to review the information architecture and whether some pages are too complex;
- Heat maps derived from eye tracking and information on the order of focus can be used to re-assess "what should be the key function of this page?"

## 6. Conclusions

Our study in this paper has provided some interesting insights into the eye fixation data from users of the BT Player app by applying machine learning and process mining technology (Markov Model). The aim of the research was to better understand how participants interacted with the app, with the ultimate goal of using these results to improve the product design in a way that will increase the sales of TV shows or films on BT Player. Also, the results have proved that the methodology of DTMC modelling and statistical analysis fits in well with industrial scenarios and requirements.

Some of these insights can demonstrate that a product design is either working, in which case it should remain the same going forward, or that there are some elements that could be improved. Such information was revealed by DTMC transition matrices as well as "steady states distribution," etc. Moreover, observing the trajectory of eye fixations over a period of time has also provided some interesting results in terms of human behaviour when viewing screens. There is evidence to corroborate the fact that we read on screens from left to right, as we would expect due to this being the same direction that we read and write ordinarily. We also read in the "F" shape, in straight lines horizontally and vertically rather than diagonally. The "most likely trajectory" on the screen was also a significant finding in our study as a summary of the possible path of eye gaze movement. Additionally, the information in relation to the expected time (mean passage time) to different screens and AOIs can be interpreted as the level of importance among different on-screen parts for viewers.

We plan to introduce more variables into the current methodology of the paper as part of our future study. For instance, the fixation duration on each AOI was not used in the paper. In that case, a continuous-time Markov chain could be applied. Also, there are some limitations to our study, one of which is that we didn't do much quantitative analysis of the eye tracker data, and that can lead to partially missing information in the human-computer interaction. Then, in addition to eye tracking technology, another method of monitoring IPTV viewer interaction should be used, such as click events on the remote control, which will provide a comparison between two parallel actions and generate a more comprehensive model.

The workflow described in this paper and its Java software implementation were re-used in a second eye-tracking study at BT, which addressed a different set of IPTV user scenarios. As such, it is believed that the methodology in this study can be a good foundation for further development and can be applied to other HCI research, e.g., usability evaluation, where an eye tracker is possibly in use. It is also generally applicable to data that comes in the format of sequential events, and such data can come from various devices like TVs, smartphones, tablets, IoT sensors, etc., and different domains. It also allows for the measurement and summarization of cognitive processes when people interact with screens.

**Author Contributions:** Conceptualization, Z.C., S.Z., S.M. and I.K.; methodology, Z.C., S.Z., S.M., M.M. and I.K.; software, Z.C., M.M. and F.H.; validation, Z.C., S.Z., S.M., F.H. and B.A.; formal analysis, Z.C. and F.H.; investigation, Z.C. and F.H.; resources, Z.C., M.M. and I.K.; data curation, Z.C. and M.M.; writing—original draft preparation, Z.C. and F.H.; writing—review and editing, S.Z., S.M., M.M., B.A. and I.K.; visualization, Z.C. and F.H.; supervision, S.Z. and S.M.; project administration, S.Z., S.M. and I.K.; funding acquisition, S.Z. and S.M. All authors have read and agreed to the published version of the manuscript.

**Funding:** This research received no external funding.

**Data Availability Statement:** Not applicable.

**Acknowledgments:** This research is supported by BTIIC (the BT Ireland Innovation Centre), funded by BT and Invest Northern Ireland.

**Conflicts of Interest:** The authors declare no conflict of interest. The funders had no role in the design of the study; in the collection, analyses, or interpretation of data; in the writing of the manuscript; or in the decision to publish the results.

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
