# Peer review of "Process Mining IPTV Customer Eye Gaze Movement Using Discrete-Time Markov Chainsâ€"

_algorithms, doi:10.3390/a16020082_

Round 1
Reviewer 1 Report
The presented study analyze of eye tracking data in authors previously published introductory paper. The authors demonstrate a way to integrate the layout information from the player with mining the process of customer eye movement on the screen, thereby gaining insights into user experience. The authors incorporate a discrete-time Markov 19 Chain, as the eye tracker records each gaze movement, which is a good example of discrete-time sequences. The abstract is a good description of the work. The introduction is well structured, and it covers most of the concepts investigated in the methodological part. In the introduction section, the research goals and the research subject and research questions should be defined more precisely. The introduction does not specify properly the contributions of the paper. The author must explain how his work is different than other similar papers. Research questions must be explained in more details. Abstract must focus only on the problem and will the paper will help in solving this problem. Please, clearly identify the contributions of the study. Please explain exactly what impact does this research have? I think some HCI references should be added to clearly classify the topic of the paper in the field of human computer iinteraction. Besides the mentioned research papers there are several other systems like BCIs, eye-tracking methods are applied nowadays and some cognitive aspects and research is relevant to this field. It would be good to see some sentences introducing the wide variety of applications of human-computer feedback systems like Examine the effect of different web-based media on human brainwaves; Study of algorithmic problem-solving and executive function; Assessing visual attention in children using gp3 eye tracker; EEG-based computer control Interface for brain-machine interaction and The analysis of hand gesture based cursor position control during solve an IT related task. Several other human-computer based monitoring systems is used in this field so please summarize these methods and applications in the introduction like quantitative analysis of relationship between visual attention and eye-hand coordination; control of incoming calls by a windows phone based brain computer interface; electroencephalogram-based brain-computer interface for internet of robotic things and evaluation of eye-movement metrics in a software debugging task using gp3 eye tracker. In the discussion section, these goals and research proposals should be clearly responded to in the light of the results obtained. How were you convinced of the validity and reliabilityof the system? Please describe the method of the evaluation considering the validity and reliability of the system.
Reviewer 2 Report
This paper is nicely written. The subject is up to date. Below you will find some general comments and friendly remarks to improve the paper.
Eye tracking technologies have been widely used in multiple areas for studies on HCI. The reviewed paper complements a previous study that was carried out at the BT Ireland Innovation Centre where eye tracking was used to record user interactions with a VoD application, the BT Player.
The authours have demonstrated integration of the layout information from the BT Player with mining customers' eye movements, thereby gaining insights into user experience from perspectives of both HCI and Industry. The study has also identified several potential areas for further work.
In section 2.1 I would appreciate a more developed review, to include a larger number of recent examples of interesting eye tracking applications in HCI-related areas, for example paper titled Horizontal vs. Vertical recommendation zones evaluation using behavior tracking, paper titled Deep Learning-Enhanced Framework for Performance Evaluation of a Recommending Interface with Varied Recommendation Position and Intensity Based on Eye-Tracking Equipment Data Processing, and a conference paper titled Evaluation of varying visual intensity and position of a recommendation in a recommending interface towards reducing habituation and improving sales,
In Section 3 the selection of machine learning model was not justified enough, in my opinion. The last statement "based on previous studies and their success, we choose Markov chains to model the trajectories of viewers’ gaze movement". Please elaborate on that, compare with other possible methods and explain why this one was selected in the end. Alternatively a few methods could be applied and compared.
In the Conclusions section please discuss the limitations of your study and recommended future directions of research in this area.
Finally, please provide sources for all figures.
Good luck with the paper. I will be available to review it again after corrections.
Reviewer 3 Report
This is an interesting and well written paper on interesting and up-to-date topic. The methodology sounds. In my opinion this paper might be in great interest for readers. I have following minor comments:
>>We have created a pipeline of data analysis in such similar case studies and come up with interpretations of Markov properties for specific scenarios as well as evaluation of user interface when using the BT Player on different pages. Also, we have generated a series of scripts, using Java, R Studio, MySQL, etc., and will make them open access on GitHub for other research purposes.<< Why do not make them open source right now? It is nowadays a common practice to publish source code together with a paper and include url to the manuscript. This greatly improves reliability and citations of the work.
>>BT Player Screenshot<< Please do not advertise trademarks and product. Hide names in text and in images.
Figures 20, 21 – number in matrices are too small to be readable. Please remove them all present in another form.
Figure 22 – please convert image to vector format, raster graphic is not suitable to line art.
Figure 23, b – plot is somehow blurred, please correct it.
Round 2
Reviewer 1 Report
I suggest to accept the submission.